# Efficient Regression-based Training of Normalizing Flows for Boltzmann Generators

**Danyal Rehman**[1,2,3,*] **Oscar Davis**[4], **Jiarui Lu**[1,2], **Jian Tang**[1,6], **Michael Bronstein**[4,5],
**Yoshua Bengio**[1,2], **Alexander Tong**[1,2,†] **Avishek Joey Bose**[1,4,7,†]
[1]Mila, [2]Université de Montréal, [3]Massachusetts Institute of Technology
[4]University of Oxford, [5]AITHYRA, [6]HEC Montréal, [7]Imperial College London

## Abstract

Simulation-free training frameworks have been at the forefront of the generative modelling revolution in continuous spaces, leading to large-scale diffusion and flow matching models. However, such modern generative models suffer from expensive inference, inhibiting their use in numerous scientific applications like Boltzmann Generators (BGs) for molecular conformations that require fast likelihood evaluation. In this paper, we revisit classical discrete-time normalizing flows in the context of BGs that offer efficient sampling and likelihoods, but whose training via maximum likelihood is often unstable and computationally challenging. We propose Regression Training of Normalizing Flows (RegFlow), a novel and scalable regression-based training objective that bypasses the numerical instability and computational challenge of conventional maximum likelihood training in favour of a simple $\ell_2$-regression objective. Specifically, RegFlow maps prior samples under our flow to targets computed using optimal transport couplings or a pre-trained continuous normalizing flow (CNF). To enhance numerical stability, RegFlow employs effective regularization strategies such as a forward-backward self-consistency loss that enjoys painless implementation. Empirically, we demonstrate that RegFlow unlocks a broader class of architectures that were previously intractable to train for BGs with maximum likelihood. We also show RegFlow *exceeds* the performance, computational cost, and stability of maximum likelihood training in equilibrium sampling in Cartesian coordinates of alanine dipeptide, tripeptide, and tetrapeptide, showcasing its potential in molecular systems. Code is available at: https://github.com/danyalrehman/RegFlow.

## 1 Introduction

The landscape of modern simulation-free generative models in continuous domains, such as diffusion models and flow matching, has led to state-of-the-art generative quality across a spectrum of domains (Betker et al., 2023; Brooks et al., 2024; Huguet et al., 2024; Geffner et al., 2025). Despite the scalability of simulation-free training, generating

Table 1: Overview of various generative models and their relative trade-offs with respect to the number of inference steps, ability to provide exact likelihoods, and training objective for learning.

| Method | One-step | Exact likelihood | Regression training |
|---|---|---|---|
| CNF (MLE) | ✗ | ✓ | ✗ |
| Flow Matching | ✗ | ✓ | ✓ |
| Shortcut (Frans et al., 2024) | ✓ | ✗ | ✓ |
| IMM (Zhou et al., 2025) | ✓ | ✗ | ✓ |
| NF (MLE) | ✓ | ✓ | ✗ |
| RegFlow (ours) | ✓ | ✓ | ✓ |

samples and computing model likelihoods from these model families requires computationally expensive inference—often hundreds of model calls—through the numerical simulation of the learned dynamical system. The search for efficient inference schemes has led to a new wave of approaches that seek to learn *one-step* generative models, either through distillation (Yin et al., 2024; Lu and Song, 2024; Sauer et al., 2024; Zhou et al., 2024), shortcut training (Frans et al., 2024), or Inductive Moment Matching (IMM) (Zhou et al., 2025) – methods that are able to retain the impressive sample quality of full simulation. However, many highly sensitive applications—for instance, in the natural sciences (Noé et al., 2019; Wirnsberger et al., 2020)—require more than just high-fidelity samples: they also necessitate accurate estimation of probabilistic quantities, the computation of which can be facilitated by having access to cheap and exact model likelihoods.

---

*Correspondence to: danyal.rehman@mila.quebec
†Equal advising

Consequently, for one-step generative models to successfully translate to scientific applications, they must additionally provide faithful *one-step exact likelihoods* that can be used to compute scientific quantities of interest, e.g., free energy differences (Rizzi et al., 2021), using the generated samples.

Given their intrinsic capacity to compute exact likelihoods, classical normalizing flows (NF) have remained the *de facto* method for generative modelling in scientific domains (Tabak and Vanden-Eijnden, 2010; Tabak and Turner, 2013; Dinh et al., 2016; Rezende and Mohamed, 2015). For example, in tasks such as equilibrium sampling of molecules, the seminal framework of Boltzmann Generators (Noé et al., 2019) pairs a normalizing flow with an importance sampling step. Consequently, rapid and exact likelihood evaluation is critical both for asymptotically debiasing generated samples in such high-impact applications and for refining them via annealed importance sampling (Tan et al., 2025a;b).

Historically, NFs employed in conventional generative modelling domains (such as images) are trained with the maximum likelihood estimation (MLE) objective, which has empirically lagged behind the expressiveness, scalability, and ease of training of modern continuous normalizing flows (CNFs) trained with regression-based objectives like flow matching and stochastic interpolants (Peluchetti, 2023; Liu, 2022; Lipman et al., 2023; Albergo and Vanden-Eijnden, 2023). A key driver of the gap between classical flows and CNFs can be attributed to the MLE training objective itself, which computes the change-of-variable formula for gradient ascent on the log-likelihood function with invertible architectures. As a result, architectures have to balance ease of optimization with expressivity, with highly flexible architectures being highly prone to being numerically unstable (Xu and Campbell, 2023; Andrade, 2024). For instance, in the context of Boltzmann Generators, this tension between MLE training and invertible architectures has led to BGs that use classical flows underfitting target molecular systems in comparison to BGs that employ flow matching (Klein et al., 2023). However, despite the expressive power of CNFs, inference still requires expensive numerical simulation—exact likelihood requires simulation of the divergence, a second-order derivative. This raises the natural motivating research question:

**Q**. *Does there exist a performant training recipe for BGs with classical NFs beyond MLE?*

**Present work**. In this paper, we answer in the affirmative. We investigate how to train an invertible neural network to directly match a predefined invertible function and build BGs with classical flows. We introduce REGRESSION TRAINING OF NORMALIZING FLOWS (REGFLOW), a novel regression-based training objective for classical normalizing flows that marks a significant departure from the well-established MLE training objective. Our key insight is that access to coupled samples from any invertible map is sufficient to train a generative model with a regression objective. As a result, we can train a classical flow by learning to match in $\ell_2$-regression the pre-computed noise-data pairings given by existing—both non-parametric or parametric—invertible maps. As a result, training REGFLOW provides similar benefits to NF training as flow matching does to continuous NFs but with the new unlocked benefit that inference provides exact likelihoods in a single step—i.e., without numerical simulation of the probability flow ODE and thus is significantly cheaper than a CNF.

To train BGs using REGFLOW, we propose a variety of couplings to facilitate simple and efficient training. We propose endpoint targets that are either: (**1**) outputs of a larger pretrained CNF; or (**2**) the solution to a pre-computed OT map done offline as a pre-processing step. To enhance training stability we also include a series of regularizers, and in particular, a new forward-backward self-consistency regularizer that completely removes the need for computing the computationally-expensive Jacobian determinant that is needed in MLE training. In each case, the designed targets are the result of already invertible mappings, which simplifies the learning problem for NFs and enhances training stability. Empirically, we deploy BG-based REGFLOW flows on learning equilibrium sampling for short peptides in alanine di-, tri-, and tetrapeptide, and find even previously discarded NF for BGs, such as affine coupling (Dinh et al., 2016) or neural spline flows (Durkan et al., 2019), can outperform their respective MLE-trained counterpart. In particular, we demonstrate that in scientific applications where MLE training is unsuccessful, the same BG model trained using REGFLOW provides higher fidelity proposal samples and likelihoods. Finally, we demonstrate a completely new method of performing Targeted Free Energy Perturbation (Wirnsberger et al., 2020) that avoids costly energy evaluations with REGFLOW that are not possible with MLE training of normalizing flows.

## 2 BACKGROUND AND PRELIMINARIES

**Generative models**. A generative model can be seen as an (approximate) solution to the distribution matching problem: given two distributions $p_0$ and $p_1$, the distributional matching problem seeks to find a push-forward map $f_\theta : \mathbb{R}^d \to \mathbb{R}^d$ that transports the initial distribution to the desired endpoint $p_1 = [f_\theta]_\#(p_0)$. Without loss of generality, we set $p_{\text{prior}} := p_0$ to be a tractable prior (typically

standard normal) and take $p_{\text{data}} := p_1$ the data distribution, from which we have empirical samples. We now turn our attention to solving the generative modelling problem with modelling families that admit exact log-likelihood, $\log p_\theta(x)$, where $p_\theta = [f_\theta]_\#(p_0)$, with a particular emphasis on normalizing flows (Tabak and Vanden-Eijnden, 2010; Tabak and Turner, 2013; Dinh et al., 2014; 2016; Rezende and Mohamed, 2015; Papamakarios et al., 2021).

## 2.1 Continuous normalizing flows

A CNF models the generative modelling problem as a (neural) ODE $\frac{d}{dt} f_{t,\theta}(x) = v_{t,\theta}(f_{t,\theta}(x_t))$. Here, $f_\theta : [0,1] \times \mathbb{R}^d \to \mathbb{R}^d, (t, x_0) \mapsto x_t$ is the smooth generator and forms the solution pathway to a (neural) ordinary differential equation (ODE) with initial conditions $f_0(x_0) = x_0$. Furthermore, $v_{t,\theta} : [0,1] \times \mathbb{R}^d \to \mathbb{R}^d$ is the time-dependent velocity field associated with the (flow) map that transports particles from $p_0$ to $p_1$. A CNF is an invertible map up to numerical precision, and as a result, we can compute the exact log-likelihood, $\log p_{t,\theta}(x_t)$, using the instantaneous change of variable formula for probability densities (Chen et al., 2018). The overall log-likelihood of a data sample, $x_0$, under the model can be computed as follows:

$$\log p_{1,\theta}(x_1) = \log p_0(x_0) - \int_1^0 \nabla \cdot v_{t,\theta}(x_t) dt. \tag{1}$$

Maximizing the model log-likelihood in eq. (1) offers one possible method to train CNF's but incurs costly simulation. Instead, modern scalable methods to train CNF's employ flow matching (Lipman et al., 2023; Albergo and Vanden-Eijnden, 2023; Tong et al., 2023; Liu et al., 2023), which learns $v_{t,\theta}$ by regressing against the (conditional) vector field associated with a designed target conditional flow everywhere in space and time, e.g., constant speed conditional vector fields.

**Numerical simulation**. In practice, the simulation of a CNF is conducted using a specific numerical integration scheme that can impact the likelihood estimate's fidelity in eq. (1). For instance, an Euler integrator tends to overestimate the log-likelihood (Tan et al., 2025a), and thus it is often preferable to utilize integrators with adaptive step size, such as Dormand–Prince(4)5 (Hairer et al., 1993). In applications where estimates of the log-likelihood suffice, it is possible to employ more efficient estimators such as Hutchinson's trace estimator to get an unbiased—yet higher variance—estimate of the divergence. Unfortunately, as we demonstrate in §3.1, such estimators are too high variance to be useful for importance sampling even in the simplest settings, and remain too computationally expensive and unreliable in larger scientific applications considered in this work.

**One-step maps: Shortcut models**. One way to discretize an ODE is to rely on the self-consistency property of ODEs, also exploited in consistency models (Song et al., 2023), namely that jumping $\Delta t$ in time can be constructed by following the velocity field for two half steps $(\Delta t/2)$. This is the core idea behind shortcut models (Frans et al., 2024) that are trained at various jumps by conditioning the vector field network on the desired step-size $\Delta t$. Precisely, $f^*_{\text{short},t,2\Delta t}(x_t) = f^*_t(x_t, \Delta t)/2 + f^*_t(x'_{t+\Delta t}, \Delta t)/2$, where $x'_{t+\Delta t} = x_t + f^*_t(x_t, \Delta t)\Delta t$. In their extreme, shortcut models define a one-step mapping which has been shown to generate high-quality images, but it remains an open question whether these models can reliably estimate likelihoods.

## 2.2 Normalizing flows

The generative modelling problem can also be tackled using time-agnostic generators. One such prominent example is Normalizing Flows (NFs) (Tabak and Vanden-Eijnden, 2010; Tabak and Turner, 2013; Dinh et al., 2016; Rezende and Mohamed, 2015), which parameterize diffeomorphisms (continuously differentiable bijective functions, with a continuously differentiable inverse), $f_\theta : \mathbb{R}^d \to \mathbb{R}^d$. For arbitrary invertible maps $f_\theta$, computing the change in log probability is prohibitively expensive with cost that scales with $O(d^3)$. Consequently, it is popular to build $f_\theta$ using a composition of $M$ elementary diffeomorphisms, each with an easier to compute Jacobian determinant: $f_\theta = f_{M-1} \circ \cdots \circ f_0$ (Papamakarios et al., 2021). Through function composition, simple invertible blocks can lead to flows that are universal density approximators (Teshima et al., 2020; Ishikawa et al., 2023; Kong and Chaudhuri, 2021; Zhang et al., 2020; Bose et al., 2021), and the resulting MLE objective for training is simply:

$$\log p_\theta(x_1) = \log p_0(x_0) - \sum_{i=0}^{M-1} \log \det \left| \frac{\partial f_{i,\theta}(x_i)}{\partial x_i} \right|, \quad p_0 := \mathcal{N}(0, I). \tag{2}$$

**Boltzmann Generators**. A Boltzmann Generator (BG) (Noé et al., 2019) combines a normalizing flow model, $p_\theta$, with an importance-sampling correction to produce i.i.d. samples from a target Boltzmann distribution $p_\text{target}$. The normalizing flow defines a tractable proposal density $p_\theta(x)$ from which we draw $K$ independent points $x^{(i)} \sim p_\theta, i \in [K]$. For each sample, we evaluate an *unnormalized* importance weight, which allow any observable $\phi(x)$ to be consistently estimated under the target measure $p_\text{target}$ using self-normalized importance sampling (SNIS) (Liu, 2001; Agapiou et al., 2017):

$$\mathbb{E}_{p_\text{target}}\left[\phi(x)\bar{w}(x)\right] \approx \frac{\sum_{i=1}^{K} w\left(x^{(i)}\right)\phi\left(x^{(i)}\right)}{\sum_{i=1}^{K} w\left(x^{(i)}\right)}, \quad w\left(x^{(i)}\right) = \frac{\exp\left(-\mathcal{E}(x^{(i)})/k_\text{B}T\right)}{p_\theta\left(x^{(i)}\right)}, \quad (3)$$

where $\mathcal{E}(x)$ denotes the potential energy and $k_\text{B}T$ are the Boltzmann constant and temperature respectively. The normalized weights $\bar{w}\left(x^{(i)}\right) = w\left(x^{(i)}\right)/\sum_j w\left(x^{(j)}\right)$ can also be used to resample the generated configurations, yielding unbiased i.i.d. draws from the desired Boltzmann distribution.

## 3 REGRESSION TRAINING OF NORMALIZING FLOWS

We seek to build one-step transport maps that both push forward samples $x_0 \sim p_0$ to $x_1 \sim p_1$, and also permit exact likelihood evaluation. Such a condition necessitates that this learned map is a bijective function—i.e. an invertible map—and enables us to compute the likelihood using the change of variable formula. While using an MLE objective is always a feasible solution to learn this map, it is often not a scalable solution for both CNFs and classical NFs. Beyond architectural choices and differentiating through a numerical solver, learning flows using MLE is intuitively harder as the process of learning must *simultaneously* learn the forward mapping, $f_\theta$, and the inverse mapping, $f_\theta^{-1}$, without knowledge of pairings $(x_0, x_1) \sim \pi(x_0, x_1)$ from a coupling.

To appreciate this nuance, consider the set of invertible mappings $\mathcal{I}$ and the subset of flows $\mathcal{F} \subset \mathcal{I}$, that solve the generative modelling problem. For instance, there may exist multiple ODEs (possibly infinitely many) that push forward $p_0$ to $p_1$. It is clear then that the MLE objective allows the choice of multiple equivalent solutions $f \in \mathcal{F}$. However, this is precisely what complicates learning $f_\theta$, as *certain* solutions are harder to optimize since there is no prescribed coupling $\pi(x_0, x_1)$ for noise $x_0$, and data targets $x_1$. That is to say, during MLE optimization of the flow $f_\theta$, the coupling $\pi$ evolves during training as it is learned in conjunction with the flow, which can often be a significant challenge to optimize when the pairing between noise and data is suboptimal.

**Regression objectives**. In order to depart from the MLE objective, we may simplify the learning problem by first picking a solution $f^* \in \mathcal{F}$ and fixing the coupling $\pi^*(x_0, x_1)$ induced under this choice, i.e. $p_1 = [f^*]_\#(p_0)$. Given privileged access to $f^*$, we can form a simple regression objective that approximates this in continuous time using our choice of learnable flow:

$$\mathcal{L}(\theta) = \mathbb{E}_{t, x_0, x_1, x_t}\left[\|f_{t,\theta}(x_t) - f_t^*(x_t)\|^2\right], \quad (4)$$

where $(x_0, x_1) \sim \pi^*(x_0, x_1)$ and $x_t \sim p_t(\cdot|x_0, x_1)$ is drawn from a known conditional noising kernel such as a Gaussian distribution. We note that the regression objective in eq. (4) is more general than just flows in $\mathcal{I}$, and, at optimality, the learned function behaves like $f_t^*$ on the support of $p_0$, under mild regularity conditions. We formalize this intuition more precisely in the next proposition.

> **Proposition 1.** *Suppose that $f_t^\star$ is invertible for all t, that $(f_t^\star)^{-1}$ is continuous for all t. Then, as $\mathcal{L}(\theta) \to 0$, it holds that $((f_t^\star)^{-1} \circ f_{t,\theta})(x) \to x$ for almost all (with respect to $p_0$) x.*

The proof for proposition 1 can be found in §A, and illuminates that solving the original generative modelling problem via MLE can be re-cast as a *matching* problem to a known invertible function $f^*$. Indeed, many existing generative models already fit into this general regression objective based on the choice of $f^*$, such as conditional flow matching (CFM) (Tong et al., 2023), rectified flow (Liu et al., 2023), and (perfect) shortcut models (Frans et al., 2024). This proposition also shows why these models work as generative models: they converge in probability to the prespecified map.

### 3.1 WARMUP: ONE-STEP GENERATIVE MODELS WITHOUT LIKELIHOOD

As there exist powerful one-step generative models in image applications, it is tempting to consider whether they can be used for BG applications requiring likelihoods. As a warmup, we investigate the use of current state-of-the-art one-step generative models in shortcut models (Frans et al., 2024) and Inductive Moment Matching (IMM) (Zhou et al., 2024) through a simple experiment (see §B for details).

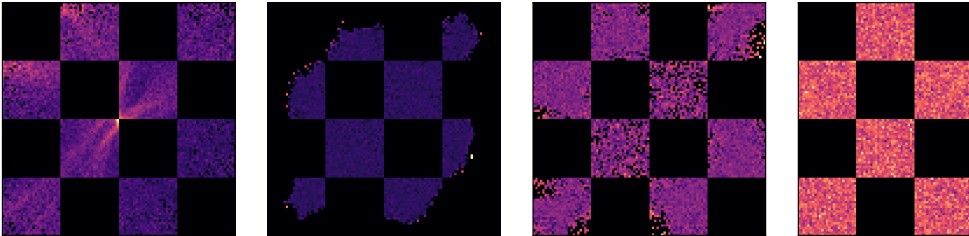

(a) Non-invertible shortcut.  (b) Non-invertible IMM.  (c) IMM with an NF.  (d) Ground truth.

Figure 1: Evaluation of IMM and shortcut models with exact likelihood on the synthetic checkerboard experiment. Depictions are provided of the 2D histograms after self-normalizing importance sampling is used.

**Synthetic experiments**. We instantiate both model classes on a simple generative modelling problem, where the data is a checkerboard density. In fig. 1, we plot the results and observe, that non-invertible shortcuts and IMM models are imperfect at learning the target and are unable to be corrected to $p_{\text{synth}}$ after resampling. However, when IMM is used to train an NF (Durkan et al., 2019), we see samples that almost perfectly match $p_{\text{synth}}$—but such an approach is not scalable (§B.2).

This puts spotlight on a counter-intuitive question given proposition 1: *Why do shortcut models have incorrect likelihoods?* While proposition 1 implies pointwise convergence of $f_\theta$ to $f^*$, this does not imply convergence or regularity of the gradients of $f_\theta$, and thus shortcut models can still achieve high quality generations without the need to provide faithful likelihoods.

**Insufficiency of uniform convergence**. One-step maps are trained to converge pointwise to $f_\theta \to f^\star$ on a sub-domain $D \subseteq \mathbb{R}^d$. However, this does not imply pointwise convergence of gradients $\nabla f_\theta \to \nabla f^\star$. For instance, consider the following toy example: $f_m(x) = \frac{1}{m} \sin(mx) + x$ and $f^\star(x) = x$. As $m \to \infty$, $f_m$ converges uniformly to $f^\star$; however, the gradient $\nabla f_m(x) = \cos(mx)$ does not converge. Importantly, this means that while $f_\theta$ would produce increasingly accurate generations, its likelihoods derived through eq. (2) may not converge to those of the base model.

## 3.2 TRAINING NORMALIZING FLOWS USING REGRESSION

We now outline our REGFLOW framework to train a one-step map for a classical NF. To remedy the issue found in shortcut models and IMM in section 3.1, we judiciously choose $f_\theta$ to be an already exactly invertible mapping—i.e., a classical NF. Since NFs are one-step maps by construction, eq. (4) is instantiated using a simple regression objective follows:

$$\mathcal{L}(\theta) = \mathbb{E}_{x_0,x_1} \left[ \|f_{1,\theta}(x_0) - f_1^*(x_0)\|^2 \right] + \lambda_r \mathcal{R} = \mathbb{E}_{x_0,x_1} \left[ \|\hat{x}_1 - x_1\|^2 \right] + \lambda_r \mathcal{R}, \quad (5)$$

where $\mathcal{R}$ is a regularization strategy and $\lambda_r \in \mathbb{R}^+$ is the strength of regularization. Explicit in eq. (5) is the need to procure *one-step* targets $x_1 = f_1^*(x_0)$ from a known invertible mapping $f_1^*$. We outline the choice of such functions in §3.3. We also highlight that the one-step targets in eq. (5) differ from the typical flow matching objective where the continuous targets $f_{t,\text{cfm}}^* = \frac{\partial}{\partial} p_t(x_t|x_0, x_1)$ (see §A.3 for a discussion). Consequently, for NFs that are universal density approximators (Teshima et al., 2020; Kong and Chaudhuri, 2021; Zhang et al., 2020), the learning problem includes a feasible solution.

**Training recipe**. We provide the full training pseudocode in algorithm 1. In practice, we find that $f^\star$ is often ill-conditioned, with the target distribution often centered around some lower-dimensional subspace of $\mathbb{R}^d$ similar to prior work (Zhai et al., 2024). This may cause $f_\theta$ to become numerically ill-conditioned. To combat this, we use three tricks to maintain numerical stability. Specifically, we regularize the loss function, add small amounts of Gaussian noise to the target distribution similar to Hui et al. (2025); Zhai et al. (2024), and, finally, add weight decay to our optimizer.

**Speedup from uni-directional flow training and inference**. For some flow types these are roughly equivalent (RealNVP or Jet) in computation time. However, for some flows such as autoregressive flows (e.g. NSF), the network $f(x)$ is substantially faster to evaluate than its inverse $f^{-1}(x)$. In standard maximum likelihood training of normalizing flows, the model is trained with passes from data to noise. This is then reversed during generation with passes from noise to data. In REGFLOW, inference *and training* can be done from noise to data. This means substantially faster inference can be achieved by training autoregressive flows where the fast direction is oriented from noise to data.

**Regularization Strategies**. In principle, classical normalizing flows can be trained using a standalone regression objective that directly maps latents to data. In practice, we observe that regression training

alone can impact numerical invertibility—a similar phenomenon to that observed in MLE-trained normalizing flows (Xu and Campbell, 2023; Andrade, 2024). This adversely impacts re-weighted samples as the NF becomes increasingly numerically unstable. To remedy this, we introduce two regularization strategies, one using the log-determinant of the Jacobian (see eq. 6), while the other does not, resembling a cycle-consistency loss using forward-backward regularization (see eq. 7):

$$\mathcal{L}_{\text{log-det}} = \|f_\theta(x_0) - x_1\|_2^2 + \lambda_r \left(\log |\det (J_\theta(x))|\right)^2 \tag{6}$$

$$\mathcal{L}_{\text{fwd-bwd}} \triangleq \|f_\theta(x_0) - x_1\|_2^2 + \lambda_r \|f_\theta^{-1}(f_\theta(x_0)) - x_0\|_2^2. \tag{7}$$

The first regularization strategy uses the same log determinant that is needed in the change of variable formula, which comes at no additional computational cost for the architectures we experiment with. Intuitively, this penalizes the flow map from collapsing to a point as it regularizes against sharp mass placements, which is what a determinant geometrically computes. The second regularizer is a new forward-backward self-consistency regularizer that ensures invertibility at the output level, but at double the computational cost. However, interestingly, since it does not require the Jacobian, it opens up potential directions for less constrained architectures. For our purposes, we find both of these regularizers accomplish our aim of avoiding collapse and maintaining invertibility.

---

**Algorithm 1** REGRESSION TRAINING OF NORMALIZING FLOWS

---

**Input:** Prior $p_0$, empirical samples from $p_1$, regularization weight $\lambda_r$, noise scale $\lambda_n$, network $f_\theta$

1: **while** training **do**
2:     $(x_0, x_1) \sim \pi(x_0, x_1)$             ▷ *Sample batches of size b i.i.d. from the dataset*
3:     $x_1 \leftarrow x_1 + \lambda_n \cdot \varepsilon$, with $\varepsilon \sim \mathcal{N}(0, I)$          ▷ *Add scaled noise to targets*
4:     $\mathcal{L}(\theta) \leftarrow \|f_\theta(x_0) - x_1\|_2^2 + \lambda_r \mathcal{R}$         ▷ *Loss with regularization*
5:     $\theta \leftarrow \text{Update}(\theta, \nabla_\theta \mathcal{L}(\theta))$
6: **return** $f_\theta$

---

### 3.3 REGFLOW TARGETS

To construct useful one-step targets in REGFLOW, we must find a discretization of an invertible function—e.g., an ODE solution—at longer time horizons. More precisely, we seek a discretization of an ODE such that each time point $t + \Delta t$ where the regression objective evaluated corresponds to a true invertible function $f_{t+\Delta t}^*$. Consequently, if we have access to an invertible map such that $t + \Delta t = 1$, we can directly regress our parametrized function as a one-step map, $f_{0,\theta}(x_0) = \hat{x}_1$. This motivates the search and design of other invertible mappings that give us invertibility at longer time horizons, for which we give two examples next.

**Optimal transport targets**. Optimal transport in continuous space between two distributions defines a continuous and invertible transformation expressible as the gradient of some convex function (Villani, 2021; Peyré and Cuturi, 2019). This allows us to consider the invertible OT plan:

$$f_{\text{ot}}^* = \arg\min_T \int T(x)c(x, T(x))dp_0(x) \text{ s.t. } T_\#(p_0) = p_1, \tag{8}$$

where $c : \mathbb{R}^d \times \mathbb{R}^d \to \mathbb{R}$ is the OT cost and $T : \mathbb{R}^d \to \mathbb{R}^d$ is a transport map. We note that this map is interesting as it requires no training; however, exact OT runs in $O(n^3)$ time and $O(n^2)$ space, which makes it challenging to scale to large datasets. Furthermore, we highlight that this differs from OT-CFM (Tong et al., 2023), which uses mini-batches to approximate the OT-plan. Nevertheless, in applicable settings, full batch OT acts as a one-time offline pre-processing step for training $f_\theta$.

**Reflow targets**. Another strategy to obtain samples from an invertible map is to use a pretrained CNF, also known as *reflow* (Liu, 2022). Specifically, we have that:

$$f_{\text{reflow}}^*(x_0) = x_0 + \int_0^1 v_t^\star(x_t)dt = x_1. \tag{9}$$

In other words, the one-step invertible map is obtained from a pre-trained CNF $v_t^\star$, from which we collect a dataset of noise-target pairs, effectively forming $\pi^*(x_0, x_1)$. We now prove that training on reflow targets with REGFLOW reduces the Wasserstein distance to the $p_1$.

**Proposition 2.** *Let $p_{reflow}$ be a pretrained CNF generated by the vector field $v_t^*$, real numbers $(L_t)_{t \in [0,1]}$ such that $v_t^*$ is $L_t$-Lipschitz for all $t \in [0,1]$, and a NF $f_\theta^{nf}$ trained using Eq. 5 by regressing against $f_{reflow}^\star(x_0)$, where $x_0 \sim \mathcal{N}(0, I)$. Then, writing $p_\theta^{nf} := \mathrm{Law}(f_\theta^{nf}(x_0))$, we have:*

$$\mathcal{W}_2(p_1, p_\theta) \leq K \exp\left(\int_0^1 L_t dt\right) + \epsilon, \quad K \geq \int_0^1 \mathbb{E}\left(\left[\|v_t^*(x_t) - v_{t,true}(x_t)\|_2^2\right]\right)^{\frac{1}{2}} dt, \quad (10)$$

*where $K$ is the $\ell_2$ approximation error between the velocity field of the CNF and the ground truth generating field $v_t^*$, $\epsilon^2 = \mathbb{E}_{x_0, x_1}\left[\|f_{reflow}^\star(x_0) - f_\theta^{nf}(x_0)\|_2^2\right]$.*

The proof for proposition 2 is provided in §A. Intuitively, the first term captures the approximation error of the pretrained CNF to the actual data distribution $p_1$, and the second term captures the approximation gap between the flow trained using REGFLOW to the reflow targets obtained via $p_{\mathrm{reflow}}$.

While these two cases represent interesting instantiations of $f^*$, there exist many other possible procedures for obtaining $f^*$. We investigate the theoretical properties for $f^*$ in appendix A.4 to provide guidance for those who wish to investigate other targets.

Table 2: Quantitative results on alanine dipeptide (ALDP), tripeptide (AL3), and tetrapeptide (AL4) reported as mean $\pm$ standard deviation over three seeds.

| Datasets → | Dipeptide (ALDP) | | | Tripeptide (AL3) | | | Tetrapeptide (AL4) | | |
|---|---|---|---|---|---|---|---|---|---|
| Algorithm ↓ | ESS ↑ | $\mathcal{E}$-$\mathcal{W}_1$ ↓ | $\mathbb{T}$-$\mathcal{W}_2$ ↓ | ESS ↑ | $\mathcal{E}$-$\mathcal{W}_1$ ↓ | $\mathbb{T}$-$\mathcal{W}_2$ ↓ | ESS ↑ | $\mathcal{E}$-$\mathcal{W}_1$ ↓ | $\mathbb{T}$-$\mathcal{W}_2$ ↓ |
| NSF (MLE) | **0.055 ± 0.012** | 13.797 ± 2.713 | 1.243 ± 0.103 | 0.024 ± 0.004 | 17.596 ± 1.21 | 1.665 ± 0.180 | **0.016 ± 0.003** | 20.886 ± 1.930 | 3.885 ± 0.410 |
| NSF (REGFLOW) | 0.035 ± 0.004 | **0.501 ± 0.011** | **0.951 ± 0.054** | **0.031 ± 0.018** | **0.853 ± 0.105** | **1.577 ± 0.140** | 0.011± 0.003 | **3.277 ± 0.546** | **2.342 ± 0.102** |
| Res–NVP (MLE) | <1e-4 | >1e3 | >30 | <1e-4 | >1e3 | >30 | <1e-4 | >1e3 | >30 |
| Res–NVP (REGFLOW) | **0.035 ± 0.008** | **2.104 ± 0.586** | **0.812 ± 0.121** | **0.025 ± 0.006** | **3.241 ± 0.301** | **1.881 ± 0.205** | **0.013 ± 0.004** | **2.705 ± 0.306** | **2.117 ± 0.331** |
| Jet (MLE) | <1e-4 | >1e3 | >30 | <1e-4 | >1e3 | >30 | <1e-4 | >1e3 | >30 |
| Jet (REGFLOW) | **0.055 ± 0.006** | **4.193 ± 1.016** | **0.801 ± 0.076** | <1e-4 | >1e3 | **3.644 ± 0.358** | <1e-4 | >1e3 | >30 |

# 4 EXPERIMENTS

We evaluate NFs trained with REGFLOW on three molecular systems: alanine dipeptide (ALDP), alanine tripeptide (AL3), and alanine tetrapeptide (AL4). These peptides are a standard benchmark for tesing generative models in computational chemistry. We asses the models on two key tasks: equilibrium conformation sampling and targeted free energy prediction (TFEP) (Wirnsberger et al., 2020). Through these experiments, we show that REGFLOW outperforms the conventional maximum likelihood estimation (MLE) training for NFs in these scientific applications.

**Setup**. We test three different architectures: RealNVP with a residual network parametrization (Dinh et al., 2016), neural spline flows (NSF) (Durkan et al., 2019), and Jet (Kolesnikov et al., 2024), across three different molecular systems (ALDP, AL3, and AL4) of increasing size and compare the performance of the same invertible architecture trained using MLE, and using REGFLOW. We report: Effective Sample Size (ESS); the 1-Wasserstein distance on the energy distribution; and the 2-Wasserstein distance on the main dihedral angles as described in §C with additional results in §D.

**Main results**. We report our main quantitative results in table 2 and observe that REGFLOW with reflow targets consistently outperforms MLE training of NFs across all architectures on both $\mathcal{E}$-$\mathcal{W}_1$ and $\mathbb{T}$-$\mathcal{W}_2$ metrics, and slightly underperforms MLE training on ESS. However, this can be justified by the mode collapse that happens in MLE training as illustrated in the Ramachandran plots for alanine dipeptide fig. 2, which artificially increases ESS. Examining the energy histogram plots in fig. 2 we observe that NFs trained using REGFLOW more closely match the true energy distribution. We also illustrate these improvements across metrics when using OT targets over reflow, as shown Appendix fig. 9. Our results clearly demonstrate that REGFLOW is often a compelling alternative to MLE training in BGs *for all analyzed NF architectures*, and allows training of architectures that were previously untrainable with MLE training.

**REGFLOW leads to faster training and inference.** We note that NFs trained with REGFLOW are far faster at computing likelihoods compared to their MLE-trained counterparts, except for cases where the NF has an analytical inverse (Res–NVP, Jet) due to the reversal of the flow. For autoregressive flows like NSF—where the reverse pass is slow to compute—we observe the maximum benefit:

Table 3: Inference efficiency comparisons. Time to compute likelihoods for 200k points.

| Models → | Dipeptide (ALDP) | | | |
|---|---|---|---|---|
| Algorithm ↓ | MLE | REGFLOW | CFM | Speed Up |
| NSF | 277.00 | 8.18 | N/A | 33.8× |
| Res–NVP | 3.64 | 3.51 | N/A | 1.03× |
| Jet | 67.63 | 60.43 | N/A | 1.11× |
| CNF DiT | N/A | N/A | 26969.80 | N/A |

Table 4: Training time comparison between MLE and REGFLOW for alanine dipeptide.

| Metric ↓ | MLE | REGFLOW | |
|---|---|---|---|
| | | OT | CNF |
| $\mathcal{E}\text{-}\mathcal{W}_1 = 7.090$ | 10h10 | 6h54 | 7h23 |
| $\mathbb{T}\text{-}\mathcal{W}_2 = 1.368$ | 12h17 | 7h32 | 7h56 |

Table 5: ALDP with various regularization strategies.

| Models → | Dipeptide (ALDP) | | |
|---|---|---|---|
| Algorithm ↓ | ESS ↑ | $\mathcal{E}\text{-}\mathcal{W}_1 \downarrow$ | $\mathbb{T}\text{-}\mathcal{W}_2 \downarrow$ |
| NSF (MLE) | $\mathbf{0.055 \pm 0.012}$ | $13.797 \pm 2.713$ | $1.243 \pm 0.103$ |
| NSF (REGFLOW w/o reg) | $0.032 \pm 0.008$ | $0.604 \pm 0.045$ | $1.083 \pm 0.109$ |
| NSF (REGFLOW w/ logdets) | $0.036 \pm 0.007$ | $0.519 \pm 0.021$ | $0.958 \pm 0.074$ |
| NSF (REGFLOW w/ fwd-bwd) | $0.035 \pm 0.004$ | $\mathbf{0.501 \pm 0.011}$ | $\mathbf{0.951 \pm 0.054}$ |
| Res–NVP (MLE) | $< 10^{-4}$ | $> 10^3$ | $> 30$ |
| Res–NVP (REGFLOW w/o reg) | $0.033 \pm 0.010$ | $2.948 \pm 0.457$ | $1.179 \pm 0.218$ |
| Res–NVP (REGFLOW w/ logdets) | $0.032 \pm 0.008$ | $2.310 \pm 0.411$ | $\mathbf{0.796 \pm 0.109}$ |
| Res–NVP (REGFLOW w/ fwd-bwd) | $\mathbf{0.035 \pm 0.008}$ | $\mathbf{2.104 \pm 0.586}$ | $0.812 \pm 0.121$ |
| Jet (MLE) | $< 10^{-4}$ | $> 10^3$ | $> 30$ |
| Jet (REGFLOW w/o reg) | $0.053 \pm 0.007$ | $9.707 \pm 1.843$ | $1.224 \pm 0.181$ |
| Jet (REGFLOW w/ logdets) | $0.051 \pm 0.004$ | $6.349 \pm 1.412$ | $0.872 \pm 0.065$ |
| Jet (REGFLOW w/ fwd-bwd) | $\mathbf{0.055 \pm 0.006}$ | $\mathbf{4.193 \pm 1.016}$ | $\mathbf{0.801 \pm 0.076}$ |

REGFLOW enables nearly a $34\times$ speedup in inference compared to the equivalent MLE-trained NF, as seen in Tab. 3. We also compare performance relative to CNFs, which require integrating the divergence of the vector field—this makes likelihood evaluation extremely expensive compared to discrete NFs. We observe that CNF inference with likelihoods is approximately $450\times$ more expensive than our slowest NF (Jet) and $7700\times$ more expensive than our fastest NF (Res–NVP).

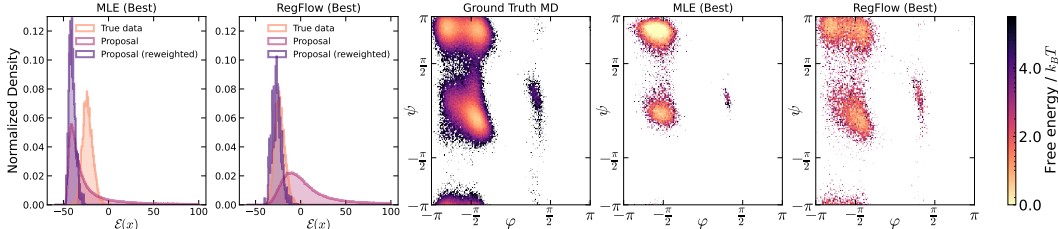

Figure 2: Energy distributions and resampled Ramachandran plots for alanine dipeptide. (**left to right**): Energy distribution of most best MLE-trained NF; energy distribution of best REGFLOW; ground truth MD data torsion angle distribution, best MLE-trained model Ramachandran plot; best REGFLOW Ramachandran plot.

Next, we contrast the training times between MLE and REGFLOW, accounting for: (1) CNF training or OT map pre-computation; (2) sample generation from the CNF; and (3) REGFLOW training until its performance exceeds MLE. Across all settings, REGFLOW consistently outperforms MLE. Specifically, we observe that achieving superior performance on $\mathcal{E}\text{-}\mathcal{W}_1$ requires $\sim$27% less time with REGFLOW, while on $\mathbb{T}\text{-}\mathcal{W}_2$, the speedup is closer to $\sim$35%. We also compare the training times between MLE and REGFLOW across all peptide systems. In fig. 3, we illustrate how the energy varies during training using REGFLOW; the dotted lines symbolize the best energy using the MLE-trained NSF on the validation set. Here, we see that the crossover between REGFLOW and MLE occurs after $\sim$1h20, $\sim$1h20, and $\sim$2h40, for REGFLOW to outperform MLE on the dipeptide, tripeptide, and tetrapeptide, respectively. Conversely, MLE training took $\sim$10h10, $\sim$11h20, and $\sim$11h40 using the dipeptide, tripeptide, and tetrapeptide, respectively. These studies further validate the potential for REGFLOW to serve as an efficient and effective alternative to MLE.

**Alternative regularization strategies.** We investigate the impact of different regularization strategies to prevent numerical collapse for REGFLOW in table 5. We consider no regularization (w/o reg), regularization of the magnitude of the log determinant of the Jacobian (w/ logdets), and a direct invertibility penalization (forward-backward). For our usecase, the Jacobian comes at no extra cost and is therefore the most efficient. The forward-backward regularizer enforces cycle consistency by performing a forward pass of the NF, followed by a reverse pass on the same generated samples, and

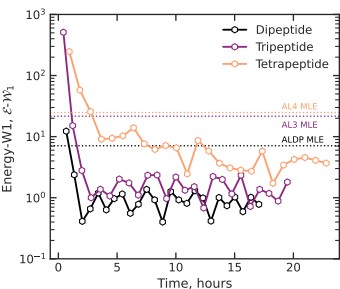

Figure 3: Training time required for REGFLOW to outperform the most performant MLE model (NSF).

computing the $\ell_2$ distance between the reconstructed priors. This is at least twice as expensive as the logdet regularization for our use case, however it does perform quite well, and interestingly opens up the possibility for more flexible architectures. All regularizations outperform MLE, and the logdet regularization offers the best tradeoff between performance and speed for our usecase, so we use that regularization for the remainder of our experiments.

**Ablations**. In table 6, we report REGFLOW using OT targets and various amounts of generated reflow targets—a unique advantage of using reflow as the invertible map. As observed, each target choice improves over MLE, outside of ESS for NSF. Importantly, we find that using more samples in reflow consistently improves performance metrics for all architectures. In fig. 4, we show how performance increases with the number reflow samples and we ablate the impact of regularization. We find performance improvements with increasing regularization, up to around $10^{-6} \leq \lambda_r \leq 10^{-5}$. Regularizing beyond this is sufficient to ensure empirical invertibility based on validation loss of $\mathcal{L}_{\text{fwd-bck}} < 10^{-4}$, but hampers generation performance.

Table 6: Ablations on target types and amount of reflow targets on ALDP.

| Datasets → | Dipeptide (ALDP) | | |
|---|---|---|---|
| Algorithm ↓ | ESS ↑ | $\mathcal{E}\text{-}\mathcal{W}_1 \downarrow$ | $\mathbb{T}\text{-}\mathcal{W}_2 \downarrow$ |
| NSF (MLE) | **0.055** | 13.80 | 1.243 |
| NSF (REGFLOW @ 100k CNF) | 0.016 | 17.39 | 1.232 |
| NSF (REGFLOW @ 10.4M CNF) | 0.035 | **0.501** | **0.951** |
| NSF (REGFLOW @ OT) | 0.003 | 0.604 | 2.019 |
| Res–NVP (MLE) | $< 10^{-4}$ | $> 10^3$ | $> 30$ |
| Res–NVP (REGFLOW @ 100k CNF) | 0.009 | 46.93 | 1.155 |
| Res–NVP (REGFLOW @ 10.4M CNF) | **0.035** | 2.104 | **0.812** |
| Res–NVP (REGFLOW @ OT) | 0.006 | **0.699** | 1.969 |
| Jet (MLE) | $< 10^{-4}$ | $> 10^3$ | $> 30$ |
| Jet (REGFLOW @ 100k CNF) | 0.017 | 31.42 | 1.081 |
| Jet (REGFLOW @ 10.4M CNF) | **0.051** | 4.193 | **0.801** |
| Jet (REGFLOW @ OT) | 0.003 | **2.534** | 1.913 |

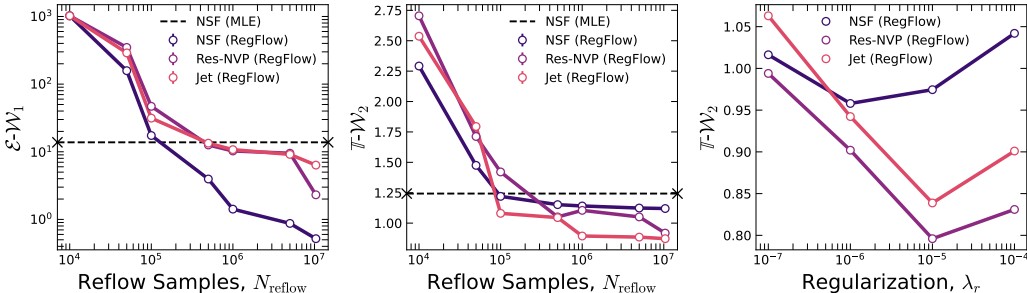

Figure 4: **Left and center**: Ablations demonstrating performance improvements with an increasing number of reflow samples. **Right**: Increasing regularization improves $\mathbb{T}\text{-}\mathcal{W}_2$ up to a certain point, beyond which numerical invertibility is guaranteed but the regression objective, and subsequently, sample quality, is adversely impacted.

**Targeted Free Energy Perturbation**. Accurate calculations of the free energy difference between two metastable states of a physical system is both ubiquitous and of profound importance in the natural sciences. One approach to tackling this problem is Free Energy Perturbation (FEP) which exploits Zwanzig's identity: $\mathbb{E}_A \left[ e^{-\beta \Delta U} \right] = e^{-\beta \Delta F}$, where $\Delta F = F_B - F_A$ is the Helmholtz free energy difference between two metastable states $A$ and $B$ (Zwanzig, 1954). Targeted Free Energy Perturbation (TFEP) improves over FEP by using NFs to learn an invertible map using MLE to increase the distributional overlap between states $A$ and $B$ (Wirnsberger et al., 2020; Moqvist et al., 2025); however, this can be challenging for several reasons. NFs are difficult to learn, especially when the energy function is expensive to compute, or the states occupy small areas.

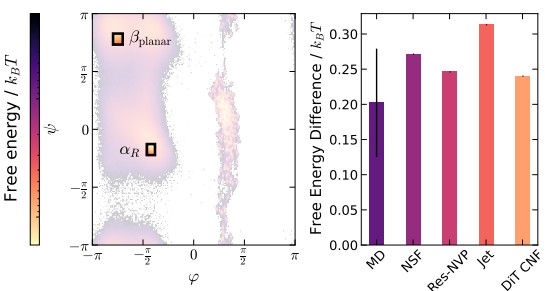

Figure 5: **Left**: The $\beta_{\text{planar}}$ and $\alpha_{\text{R}}$ conformation states; **Right**: REGFLOW's ability to learn free energy differences.

We propose a new TFEP method that does not require energy function evaluations during training. By using REGFLOW, we can train the NF solely based on samples from states $A$ and $B$. This enables TFEP, where energy evaluations may be costly—a new possibility that is distinct from NFs trained using MLE. To demonstrate this application of REGFLOW, we train an NF solely from samples from two modes of ALDP (see fig. 5) and use OT targets which avoid *any energy function evaluation*.

We include a reference using the DiT CNF—trained to map between meta-stable states—which also achieves similar predictions, albeit taking nearly three orders of magnitude longer to compute. We find we can achieve high-quality free energy estimation in comparison to ground truth Molecular Dynamics (MD) using only samples during training, as illustrated in fig. 5. We believe this is a promising direction for future applications of free energy prediction.

## 5 RELATED WORK

**Exact likelihood generative models**. NFs are generative models with invertible architectures (Rezende and Mohamed, 2015; Dinh et al., 2016) that produce *exact* likelihoods for any given points. Common models include RealNVP (Dinh et al., 2016), neural spline flows (Durkan et al., 2019), and Glow (Kingma and Dhariwal, 2018). Jet (Kolesnikov et al., 2024) and TarFlow (Zhai et al., 2024) are examples of transformer-based normalizing flows. Aside from Jet and Tarflow, NFs have generally underperformed compared to diffusion models and flow matching methods (Ho et al., 2020; Lipman et al., 2023; Albergo et al., 2023; Liu, 2022), partly due to the high computational cost of evaluating the log-determinants of Jacobians at each training step.

**Few-step generative models**. To avoid costly inference, few-step generative models were introduced as methods to accelerate the simulation of diffusion and CNFs. Common examples include DDIM (Song et al., 2022) and consistency models (Song et al., 2023), which introduced a new training procedure that ensured the model's endpoint prediction remained consistent. Recently, flow maps (Boffi et al., 2024; 2025; Song and Dhariwal, 2023; Lu and Song, 2024; Geng et al., 2024; 2025; Sabour et al., 2025) have improved upon this paradigm. Other lines of work proposed related but different training objectives, generalizing consistency training (Frans et al., 2024; Zhou et al., 2025; Kim et al., 2024; Heek et al., 2024). Beyond diffusion and FM, residual networks (He et al., 2015) are a class of neural networks that are invertible if the Lipschitz constant of $f_\theta$ is at most one (Behrmann et al., 2019). The log-determinant of the Jacobian is then approximated by truncating a series of traces (Behrmann et al., 2019)—an approximation improved in Chen et al. (2020).

## 6 CONCLUSION

In this work, we present REGFLOW, a method for generating high-quality samples alongside exact likelihoods in a single step. Using a base coupling between the dataset samples and the prior, provided by either pre-computed optimal transport or a base CNF, we can train a classical NF using a simple regression objective that avoids computing Jacobians at training time, as opposed to typical MLE training. In theory and practice, we have shown that the learned model produces faithful samples, the likelihoods of which empirically allow us to produce state-of-the-art results on several molecular datasets, using importance-sampling resampling. Limitations include the quality of the proposal samples, which substantially improve on MLE-trained NFs, but are not on par with state-of-the-art CNFs or variants thereof. Moreover, while producing accurate and high-quality likelihoods, they do not, in theory, match those of the base coupling, which can be a desirable property.

## 7    ACKNOWLEDGEMENTS

DR received financial support from the Natural Sciences and Engineering Research Council's (NSERC) Banting Postdoctoral Fellowship under Funding Reference No. 198506. OD is supported by both Project CETI and Intel. AJB is partially supported by an NSERC Postdoctoral Fellowship and by the EPSRC Turing AI World-Leading Research Fellowship No. EP/X040062/1 and EPSRC AI Hub No. EP/Y028872/1. JT acknowledges funding from the Canada CIFAR AI Chair Program and the Intel-Mila partnership program. The authors acknowledge funding from UNIQUE, CIFAR, NSERC, Intel, and Samsung. The research was enabled in part by computational resources provided by the Digital Research Alliance of Canada (https://alliancecan.ca), Mila (https://mila.quebec), and NVIDIA.

## ETHICS STATEMENT

This paper is primarily methodological, presenting theoretical developments without direct experimental implementation or associated ethical considerations; however, we advise due caution for future beneficiaries of our work in their potentially sensitive application domains.

## REPRODUCIBILITY STATEMENT

We have made numerous efforts to ensure the reproducibility of our work. The main paper provides detailed descriptions of the proposed methods and evaluation protocols. Additionally, in our Appendix, we include extensive details on data normalization, the MD datasets used for training, model architectures and sizes, training configurations, and our choice of regularization hyperparameters. Further, all assumptions and methodological choices are explicitly documented, and we plan to publicly release all the developed code upon acceptance.

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

# A  PROOFS

## A.1  PROOF OF PROPOSITION 1

We first recall proposition 1 below.

> **Proposition 1.** *Suppose that $f_t^\star$ is invertible for all $t$, that $(f_t^\star)^{-1}$ is continuous for all $t$. Then, as $\mathcal{L}(\theta) \to 0$, it holds that $((f_t^\star)^{-1} \circ f_{t,\theta})(x) \to x$ for almost all (with respect to $p_0$) $x$.*

To prove proposition 1, we first prove the following lemma, which is essentially the same as the proposition, but it abstracts out the distribution of $x_t$, which depends on $x_0$, $x_1$, and $t$.

> **Lemma 1.** *For functions $(f_n)_{n\geq 1}$ and $g$, where $g$ is invertible and has a continuous inverse, $x_0 \sim p_0$, if $\mathrm{MSE}(f_n, g) := \mathbb{E}_{x_0} \|f_n(x_0) - g(x_0)\|_2^2 \to 0$, then $\lim_{n\to\infty} g^{-1}(f_n(x)) = x$ for almost all (with respect to $p_0$) $x$.*

*Proof.* Let $Y_n = \|f_n(x_0) - g(x_0)\|_2$. We know that $\lim_{n\to\infty} \mathbb{E}[Y_n^2] = 0$ (as it corresponds to the MSE), which implies that $\lim_{n\to\infty} \mathrm{Var}(Y_n) = 0$. Consequently, $Y_n \to c$ for some constant $c \in \mathbb{R}$. Moreover, by Jensen's inequality and the convexity of $x \mapsto x^2$, we find that $(\mathbb{E}[Y_n])^2 \leq \mathbb{E}[Y_n^2]$, meaning that $c = 0$. This implies that $\lim_{n\to\infty} \|f_n(x) - g(x)\|_2^2 = 0$ almost everywhere, and thus that $\lim_{n\to\infty} f_n(x) = g(x)$. Finally, since $g^{-1}$ is continuous, we can apply the function to both sides of the limit to find that $\lim_{n\to\infty} g^{-1}(f_n(x)) = x$, almost everywhere. $\qquad\square$

It suffices to apply the above lemma to $x_t \sim p_t(\,\cdot\, | x_0, x_1)p_1(x_1 | x_0)p_0(x_0)$.

## A.2  PROOF OF PROPOSITION 2

We now prove proposition 2. The proposition reuses the following regularity assumptions, as introduced in Benton et al. (2023), which we recall verbatim below for convenience:

**(Assumption 1)** Let $v_{\text{true}}$ be the true generating velocity field for the CNF with field $v^*$ trained using flow matching. Then the true and learned velocity $v^*$ are close in $\ell_2$ and satisfy: $\int_0^1 \mathbb{E}_{t,x_t}[\|v_{t,\text{true}}(x_t) - v_t^*(x_t)\|^2]dt \leq K^2$.

**(Assumption 2)** For each $x \in \mathbb{R}^d$ and $s \in [0,1]$, there exists unique flows $(f_{s,t}^*)_{t\in[s,1]}$ and $(f_{(s,t),\text{true}})_{t\in[s,1]}$, starting at $f_{(s,s)}^* = x$ and $f_{(s,s),\text{true}} = x$ with velocity fields $v_t^*(x_t)$ and $v_{t,\text{true}}(x_t)$, respectively. Additionally, $f^*$ and $f_{\text{true}}$ are continuously differentiable in $x$, $s$ and $t$.

**(Assumption 3)** The velocity field $v_t^*(x_t)$ is differentiable in both $x$ and $t$, and also for each $t \in [0,1]$ there exists a constant $L_t$ such that $v_t^*(x_t)$ is $L_t$-Lipschitz in $x$.

> **Proposition 2.** *Let $p_{reflow}$ be a pretrained CNF generated by the vector field $v_t^*$, real numbers $(L_t)_{t\in[0,1]}$ such that $v_t^*$ is $L_t$-Lipschitz for all $t \in [0,1]$, and a NF $f_\theta^{nf}$ trained using Eq. 5 by regressing against $f_{reflow}^\star(x_0)$, where $x_0 \sim \mathcal{N}(0, I)$. Then, writing $p_\theta^{nf} := \mathrm{Law}(f_\theta^{nf}(x_0))$, we have:*
>
> $$\mathcal{W}_2(p_1, p_\theta) \leq K \exp\left(\int_0^1 L_t dt\right) + \epsilon, \quad K \geq \int_0^1 \mathbb{E}\left(\left[\|v_t^*(x_t) - v_{t,true}(x_t)\|_2^2\right]\right)^{\frac{1}{2}} dt, \quad (10)$$
>
> *where $K$ is the $\ell_2$ approximation error between the velocity field of the CNF and the ground truth generating field $v_t^*$, $\epsilon^2 = \mathbb{E}_{x_0,x_1}\left[\|f_{reflow}^\star(x_0) - f_\theta^{nf}(x_0)\|_2^2\right]$.*

*Proof.* We begin by first applying the triangle inequality to $\mathcal{W}_2(p_1, p_\theta)$ and obtain:

$$\mathcal{W}_2(p_1, p_\theta) \leq \mathcal{W}_2(p_1, p_{\text{reflow}}) + \mathcal{W}_2(p_{\text{reflow}}, p_\theta^{\text{nf}}). \quad (11)$$

The first term is an error in Wasserstein-2 distance between the true data distribution and our reflow targets, which is still a CNF. A straightforward application of Theorem 1 in Benton et al. (2023) gives a bound on this first Wasserstein-2 distance[1]:

$$\mathcal{W}_2(p_1, p_{\text{reflow}}) \leq K \exp\left(\int_0^1 L_t dt\right). \quad (12)$$

---

[1] A sharper bound can be obtained with additional assumptions, as demonstrated in Benton et al. (2023), but it is not critically important in our context.

To bound $\mathcal{W}_2(p_{\text{reflow}}, p_\theta)$, recall that the following inequality holds $\mathcal{W}_2(\text{Law}(X), \text{Law}(Y)) \leq \mathbb{E}\left[\|X - Y\|_2^2\right]^{\frac{1}{2}}$, for any two random variables $X$ and $Y$. In our case, these random variables are $p_{\text{reflow}}^* = \text{Law}(f_{\text{reflow}}^*(x_0))$ and $p_\theta^{\text{nf}} = \text{Law}(f_\theta^{\text{nf}}(x_0))$. This gives:

$$\mathcal{W}_2(p_{\text{reflow}}, p_\theta^{\text{nf}}) \leq \mathbb{E}_{x_0, x_1}\left[\left\|f_{\text{reflow}}^*(x_0) - f_\theta^{\text{nf}}(x_0)\right\|_2^2\right]^{\frac{1}{2}}. \tag{13}$$

Combining eq. (12) and eq. (13) achieves the desired result and completes the proof.

$$\mathcal{W}_2(p_1, p_\theta) \leq K \exp\left(\int_0^1 L_t dt\right) + \mathbb{E}_{x_0, x_1}\left[\left\|f_{\text{reflow}}^*(x_0) - f_\theta^{\text{nf}}(x_0)\right\|_2^2\right]^{\frac{1}{2}}. \tag{14}$$

$\square$

Note that the bound on $\mathcal{W}_2(p_{\text{reflow}}, p_\theta^{\text{nf}})$ is effectively the square-root of the REGFLOW objective and thus optimization of the NF using this loss directly minimizes the upper bound to $\mathcal{W}_2(p_1, p_\theta^{\text{nf}})$.

### A.3 REGFLOW IN CONTINUOUS TIME

Current state-of-the-art CNFs are trained using "flow matching" (Lipman et al., 2023; Albergo and Vanden-Eijnden, 2023; Liu et al., 2023), which attempts to match the vector field associated with the flow to a target vector field that solves for mass transportation everywhere in space and time. Specifically, we can cast conditional flow matching (CFM) (Tong et al., 2023) from the perspective of REGFLOW. To see this explicitly, consider a pre-specified probability path, $p_t(x_t)$, and the following $f_{t,\text{fm}}^* = \frac{\partial}{\partial t} p_t(x_t)$. However, since it is generally computationally challenging to sample from $p_t$ directly, the marginalization trick is used to derive an equivalent objective with a conditional $f_{t,\text{cfm}}^*$. We note that REGFLOW requires $f_{t,\text{cfm}}^*$ to be invertible therefore this assumes regularity on $\frac{\partial}{\partial t} p_t(x_t)$. This is generally satisfied by adding a small amount of noise to the following. We present this simplified form for clarity.

$$p_t(x_t) := \int p_t(x_t|x_0, x_1) d\pi(x_0, x_1), \quad p_t(x_t|x_0, x_1) = \delta(x_t; (1-t)x_0 + tx_1). \tag{15}$$

Then setting $f_{t,\text{cfm}}^* = \frac{\partial}{\partial t} p_t(x_t|x_0, x_1)$ it is easy to show that:

$$\mathcal{L}(\theta) = \mathbb{E}_{t, x_0, x_1, x_t}\left[\left\|v_{t,\theta}(x_t) - \frac{\partial}{\partial t} p_t(x_t|x_0, x_1)\right\|^2\right] = \mathbb{E}_{t, x_t}\left[\left\|v_{t,\theta}(x_t) - \frac{\partial}{\partial t} p_t(x_t)\right\|^2\right] + C,$$

$$= \mathbb{E}_{t, x_0, x_1, x_t}\left[\lambda_t \left\|f_{t,\theta}(x_t) - f_{t,\text{cfm}}^*(x_t)\right\|^2\right],$$

with $C$ independent of $\theta$ (Lipman et al., 2023), and $\lambda_t$ is a loss weighting, which fits within the REGFLOW framework in the continuous-time setting with the last equality known as target/end-point prediction.

### A.4 REQUIREMENTS FOR REGFLOW TARGETS

In practice, REGFLOW deals with discrete couplings. Any discrete coupling is usable for training RegFlow as long as there exists an invertible function on the continuous domain which agrees with it. This leaves us with easily verifiable necessary and sufficient properties for the base coupling. Specifically, let $\pi(x_0, x_1)$ denote the coupling between empirical point sets $x_0, x_1$ in $\mathbb{R}^d$.

**Proposition 3.** *If $\pi$ is a permutation and there does not exist $i \neq j$ such that $x_0^i = x_0^j$ or $x_1^i = x_1^j$. Then there exists invertible function $f : \mathbb{R}^d \to \mathbb{R}^d$ such that for all $x_0^i \in x_0$, $f(x_0^i) = x_1^i$ and for all $x_1^i \in x_1$, $f^{-1}(x_1^i) = x_0^i$.*

*Proof.* We proceed by constructing an example $f$ which satisfies the necessary properties. First we denote $f^* : x_0 \to x_1$ as the discrete invertible function mapping the point set $x_0$ to the point set $x_1$. Let

$$f(x) = \begin{cases} f^*(x) & \text{if } x \in x_0 \\ f^{*-1}(x) & \text{if } x \in x_1 \\ x & \text{else} \end{cases}$$

This function is invertible on $\mathbb{R}^d$ and satisfies the necessary properties in the proposition on the domains of $x_0$ and $x_1$. $\square$

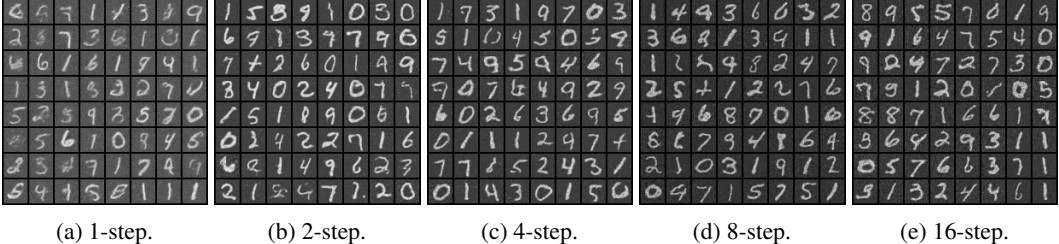

| (a) 1-step. | (b) 2-step. | (c) 4-step. | (d) 8-step. | (e) 16-step. |

Figure 6: Generations of IMM trained with an iUNet with a variable number of steps.

This proposition establishes necessary and sufficient conditions for training a valid REGFLOW between $x_0$ and $x_1$. This property is quite simple to obtain in practice. For both OT-couplings and CNF-couplings $\pi$ is almost by definition a permutation. The only trouble is if there exist duplicate points. This is a measure-zero event in continuous space, and therefore is not an issue.

In fact, any random permutation matrix satisfies these conditions. Which leads to the perhaps more interesting question of what are the properties of a "good" coupling $\pi$. In some sense we are looking for $\pi$ that are "good" couplings that are somehow "easy" to learn and generalizes well when trained with the REGFLOW procedure in a given setting.

In this work we used REGFLOW to improve the training speed and convergence of the same normalizing flow architectures that are normally trained using maximum likelihood (MLE).

We believe the classic MLE objective would help when it is difficult to find a good coupling for the given architecture, dataset, and REGFLOW learning framework. In this work we established that there exist settings where OT and CNF-couplings outperform the MLE objective. We leave it to future work to study the optimal couplings in a given setting.

## B  ADDITIONAL BACKGROUND

### B.1  INDUCTIVE MOMENT MATCHING

Introduced in Zhou et al. (2025), Inductive Moment Matching (IMM) defines a training procedure for one-step generative models, based on diffusion/flow matching. Specifically, IMM trains models to minimize the difference in distribution between different points in time induced by the model. As a result, this avoids direct optimization for the predicted endpoint, in contrast to conventional diffusion.

More precisely, let $f_\theta : \mathbb{R}^d \times [0, 1]^2 \to \mathbb{R}^d, (x, s, t) \mapsto f_\theta(x, s, t)$ be a function parameterized by $\theta$. IMM minimizes the following maximum mean discrepancy (MMD) loss:

$$\mathcal{L}(\theta_n) = \mathbb{E}_{s,t,x_0,x_1} \left[ w(s,t)\text{MMD}^2 \left( p_{\theta_{n-1},(s|r)}(x_s), p_{\theta_n,(s|t)}(x_s) \right) \right], \tag{16}$$

where $0 \le r \le r(s,t) := r \le s \le 1$, with $s, t \sim \mathcal{U}(0,1)$ iid, $w \ge 0$ is a weighting function, $x_1$ is a sample from the target distribution, $x_0 \sim \mathcal{N}(0, I)$, $x_s$ is some interpolation between $x_0$ and $x_1$ at time $s$ (typically, using the DDIM interpolation (Song et al., 2022)), the subscript $n \in \mathbb{N}$ of parameter $\theta$ refers to its training step, and MMD is some MMD function based on a chosen kernel (typically, Laplace).[2] Essentially, the method uses as a target the learned distribution of the previous step at a higher time to train the current distribution at lower times. With a skip parameterization, the higher time distribution is by construction close to the true solution, as $p_\theta(x_s \mid x_r) \approx p(x_s \mid x_r)$ when $r \approx s$, and $x_s$ is known. (Or, in other terms, $f_\theta(x, s, r \approx s) \approx x$ with the skip parameterization.) When the distributions match (when the loss is zero), $\text{MMD}^2(p_{1,\theta}, p_1) = 0$, and so the generative model's and the target distribution's respective moments all match.

This training procedure allows for variable-step sampling. For chosen timesteps, $(t_i)_{i=1}^n$, one can sample from $p_{1,\theta}$ by sampling $x_0 \sim \mathcal{N}(0, I)$ and performing the steps:

$$x_{t_{i+1}} \leftarrow \text{DDIM}(f_\theta(x_{t_i}, t_{i+1}, t_i), x_{t_i}, t_i, t_{i+1}), \tag{17}$$

where DDIM is the DDIM interpolant.

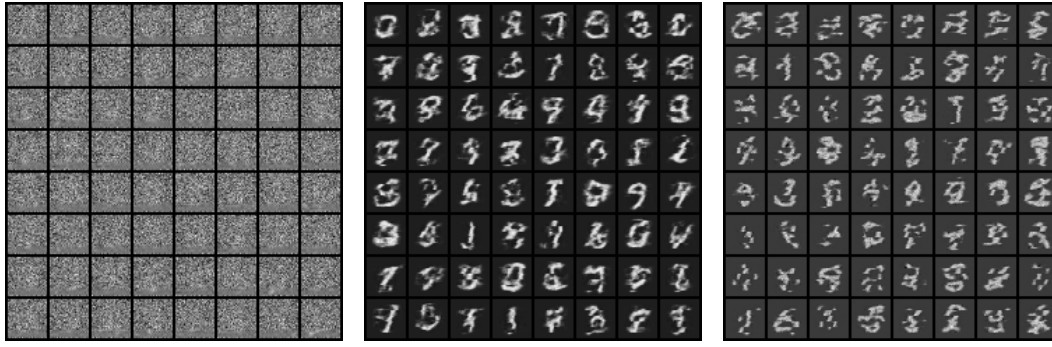

(a) Using the ResFlow architecture proposed in Chen et al. (2020).

(b) Using the TarFlow architecture (Zhai et al., 2024), $m = 4$.

(c) Using the TarFlow architecture (Zhai et al., 2024), $m = 16$.

Figure 7: One-step generation results with a Lipschitz-constrained (ResFlow) model and an invertible model (TarFlow) for IMM. The $m$ parameter is the group size in IMM used to approximate the MMD.

## B.2 INDUCTIVE MOMENT MATCHING NEGATIVE RESULTS

We detail in appendix B.1 the Inductive Moment Matching (IMM) framework (Zhou et al., 2025). Observing the sampling procedure, which we give in eq. (17), one can make this procedure invertible by constraining the Lipschitz constant of the model, or by using an invertible model. For the first case, if we use the "Euler" (skip) parameterization alongside the DDIM interpolation, it is shown that the reparameterized model $g_\theta$ can be written as:

$$\forall x, s, t, \qquad g_\theta(x, s, t) = x - (s - t)f_\theta(x, s, t). \tag{18}$$

Moreover, $0 \le s - t \le 1$, and so if the Lipschitz constant of $f_\theta$ is strictly less than one, then the overall model is invertible, using the argument of residual flows (Behrmann et al., 2019); so the change of variables formula applies as follows (using the time notation of IMM/diffusion):

$$\log p_1^\theta(x) = \log p_0(x_0) - \sum_i \log \left[(t_{i+1} - t_i) \det(J_{f_\theta(\cdot, t_{i+1}, t_i)}(x_{t_i}))\right], \tag{19}$$

The difficulty of evaluating the log-determinant of the Jacobian remains. Note, however, that we do not need to find the inverse of the function to evaluate the likelihood of generated samples, since we know each $(x_{t_i})_i$. The second path (of using an invertible model) is viable only for one-step sampling with no skip parameterization (which, according to Zhou et al. (2025), tends to under-perform, empirically), since the sampling procedure then boils down to $x_1 = f(x_0, 1, 0)$ for $x_0 \sim \mathcal{N}(0, I)$.

While both approaches succeeded in synthetic experiments, they fail to scale to datasets such as MNIST, the results of which we include here in fig. 6 and in fig. 7. We have tried iUNet (Etmann et al., 2020) and TarFlow (Zhai et al., 2024), an invertible UNet and a Transformer-based normalizing flow, respectively, for invertible one-step models; and we have tried the ResFlow architecture in (Chen et al., 2020) for the Lipschitz-constrained approach. As observed, TarFlow fails to produce images of high quality; iUNets produced significantly better results, albeit still not sufficient, especially for the one-step sampling, which is the only configuration that guarantees invertibility; the Lipschitz-constrained ResFlow entirely failed to produce satisfactory results, although the loss did diminish during training. In general, an even more important limitation is the difficulty of designing invertible or Lipschitz-constrained models for other data types, for instance, 3D coordinates. Perhaps further research on the architectural side could allow for higher performance with invertible sampling.

## C EXPERIMENTAL DETAILS

### C.1 METRICS

The performance metrics considered across the investigated flows were the effective sample size, ESS, Wasserstein-1 energy distance, $\mathcal{E}\text{-}\mathcal{W}_1$, and the Wasserstein-2 distance on dihedral angles, $\mathbb{T}\text{-}\mathcal{W}_2$.

---

[2]Note that we have adapted IMM's notation to our time notation, with noise at time zero, and clean data at time one.

**Effective Sample Size (ESS)**. We compute the effective sample size (ESS) using Kish's formula, normalized by the number of samples generated:

$$\text{ESS}\left(\{w_i\}_{i=1}^N\right) = \frac{1}{N} \frac{\left(\sum_{i=1}^N w_i\right)^2}{\sum_{i=1}^N w_i^2}. \tag{20}$$

where $w_i$ is the unnormalized weight of each particle indexed by $i$ over $N$ particles. Effective sample size measures the variance of the weights and approximately how many more samples would be needed compared to an unbiased sample. For us, this captures the local quality of the proposal relative to the ground truth energy. It does not rely on a ground truth test set; however, it is quite sensitive and may be misleading in the case of dropped modes or incomplete coverage, as it only measures agreement on the support of the generated distribution.

**Wasserstein-1 Energy Distance ($\mathcal{E}$-$\mathcal{W}_1$)**. The Wasserstein-1 energy distance measures how well the generated distribution matches some ground truth sample (often generated using MD data) by calculating the Wasserstein-1 distance between the energy histograms. Specifically:

$$\mathcal{E}\text{-}\mathcal{W}_1(x, y) = \min_\pi \int_{x,y} |x - y| d\pi(x, y), \tag{21}$$

where $\pi$ is a valid coupling of $p(x)$ and $p(y)$. For discrete distributions of equal size, $\pi$ can be thought of as a permutation matrix. This measures the model's ability to generate very accurate structures as the energy function we use requires extremely accurate bond lengths to obtain reasonable energy values. When the bond lengths have minor inaccuracies, the energy can blow up extremely quickly.

**Torus Wasserstein ($\mathbb{T}$-$\mathcal{W}_2$)**. The torus Wasserstein distance measures the Wasserstein-2 distance on the torus defined by the main torsion angles of the peptide. That is for a peptide of length $l$, there are $2(l-1)$ torsion angles defining the *dihedrals* along the backbone of interest $((\phi_1, \psi_1), (\phi_2, \psi_2), \ldots (\phi_l, \psi_l))$. We define the torus Wasserstein distance over these backbone angles as:

$$\mathbb{T}\text{-}\mathcal{W}_2(p, q)^2 = \min_\pi \int_{x,y} c_{\mathcal{T}}(x, y)^2 d\pi(x, y), \tag{22}$$

where $\pi$ is a valid coupling between $p$ and $q$, and $c_{\mathcal{T}}(x, y)^2$ is the shortest distance on the torus defined by the dihedral angles:

$$c_{\mathcal{T}}(x, y)^2 = \sum_{i=0}^{2(L-1)} \left[(\text{Dihedrals}(x)_i - \text{Dihedrals}(y)_i + \pi) \bmod 2\pi - \pi\right]^2. \tag{23}$$

The torus Wasserstein distance measures large scale changes and is quite important for understanding mode coverage and overall macro distribution. We find REGFLOW does quite well in this regard.

## C.2 ADDITIONAL DETAILS ON EXPERIMENTAL SETUP

To accurately compute the previously defined metrics, 250k proposal samples were drawn and re-weighted for alanine dipeptide, tripeptide, and tetrapeptide.

**Data normalization**. We adopt the same data normalization strategy proposed in (Tan et al., 2025a), in which the center of mass of each atom is first subtracted from the data, followed by scaling using the standard deviation of the training set.

**Exponential moving average**. We apply an exponential moving average (EMA) on the weights of all models, with a decay of 0.999, as commonly done in flow-based approaches to improve performance.

**Training details and hardware**. All models were trained on NVIDIA L40S 48GB GPUs for 5000 epochs, except those using OT targets, which were trained for 2000 epochs. Convergence was noted earlier in the OT experiments, leading to early stopping. The total training time for all models is summarized in table 7. The time taken to compute the OT map is also provided; since computing the OT map is independent of the feature dimension, but only on the number of data points used, the compute time was relatively consistent across all datasets. A total of 100k points was used for training the CNF, performing MLE training, and computing the OT map.

Table 7: REGFLOW training time (in hours) on ALDP, AL3, and AL4.

| Model | ALDP | AL3 | AL4 |
|---|---|---|---|
| OT map | 3.6 | 3.8 | 3.8 |
| DiT CNF | 27.6 | 40.7 | 48.6 |
| NSF | 21.0 | 23.8 | 26.8 |
| Res–NVP | 15.7 | 15.6 | 15.0 |
| Jet | 19.1 | 19.2 | 20.1 |

**Reflow targets**. Ablations were done to investigate the influence of synthetic data quantity on all metrics. For all benchmarking performed against MLE training, the largest amount of synthetic data was used. For ALDP, AL3, and AL4, this constituted 10.4M, 10.4M, and 10M samples, respectively.

**Determinant regularization**. During REGFLOW, it was initially observed that as proposal sample quality improved, the re-weighted samples progressively deteriorated across all metrics due to the models becoming numerically non-invertible. This was partially addressed by adding regularization to the loss in the form of a log determinant penalty. Sweeps were conducted using multiple regularization weights ranging between $10^{-7}$ and $10^{-4}$ to prevent hampering sample performance. The amount of regularization added was a function of the flow and dataset. The final weights are summarized in table 8.

Table 8: Regularization weights used across datasets and flows.

| Model | ALDP | AL3 | AL4 |
|---|---|---|---|
| NSF | $10^{-6}$ | $10^{-5}$ | $10^{-5}$ |
| Res–NVP | $10^{-5}$ | $10^{-5}$ | $10^{-6}$ |
| Jet | $10^{-5}$ | $10^{-6}$ | $10^{-5}$ |

**Target noise**. To discourage numerical non-invertibility of the trained flows, Guassian noise was also introduced to the target samples. Experiments were conducted with noise magnitudes of 0.01, 0.05, 0.1, and 0.25, with a final value of 0.05 being selected for use across models and datasets.

**REGFLOW implementation details**. A summary of all trained model configurations is provided in table 9. To maintain a fair comparison, the configurations reported below were unchanged for MLE training and REGFLOW. Adam was used as the optimizer with a learning rate of $5 \times 10^{-4}$ and a weight decay of 0.01. We also included a varying cosine schedule with warmup in line with the approach suggested in (Tan et al., 2025a).

Table 9: Model configurations for the DiT CNF, NSF, Res–NVP, and Jet across all datasets (ALDP, AL3, AL4). A dash (–) indicates the parameter is not applicable to the respective model.

| Model | hidden features | transforms | layers | blocks per layer | conditioning dim. | heads | dropout | # parameters (M) |
|---|---|---|---|---|---|---|---|---|
| DiT CNF | 768 | – | 6 | – | 128 | 12 | 0.1 | 46.3 |
| NSF | 256 | 24 | – | 5 | – | – | – | 76.8 |
| Res–NVP | 512 | – | 8 | 6 | – | – | 0.1 | 80.6 |
| Jet | 432 | – | 4 | 12 | 128 | 12 | 0.1 | 77.6 |

**Quality of CNF targets**. To maximize the likelihood that models trained with REGFLOW have the potential to outperform MLE, securing high-quality targets is essential. In line with this pursuit, a CNF with a diffusion transformer backbone was used. In fig. 8, the true data and the CNF proposal are shown, where it can be seen that the learned energy distributions across all three peptides are nearly perfect. Re-weighted samples are not included as obtaining likelihoods from the CNF requires estimating the trace of the divergence, which is often an expensive operation with a large time and memory cost. Although many unbiased approaches for approximating the likelihood exist (Hutchinson, 1989), these methods are typically unusable for Boltzmann Generators due to their variance, which can introduce bias into the weights needed for importance sampling.

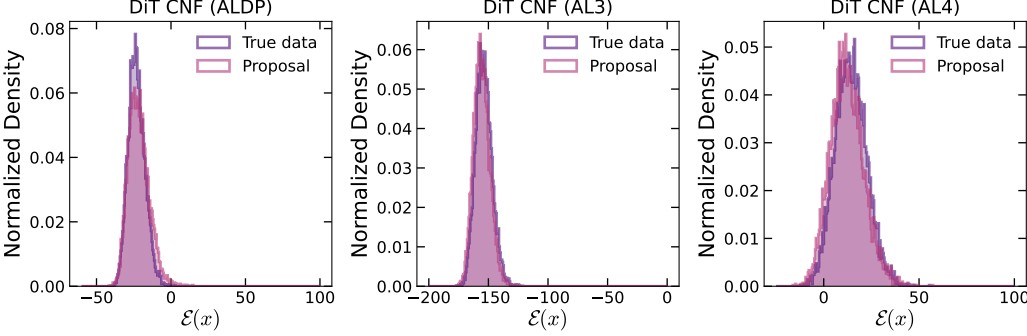

Figure 8: True energy distribution and learned proposal using the DiT-based CNF. *The re-weighted proposal is not present because it was too computationally expensive to compute for a sufficient number of points.

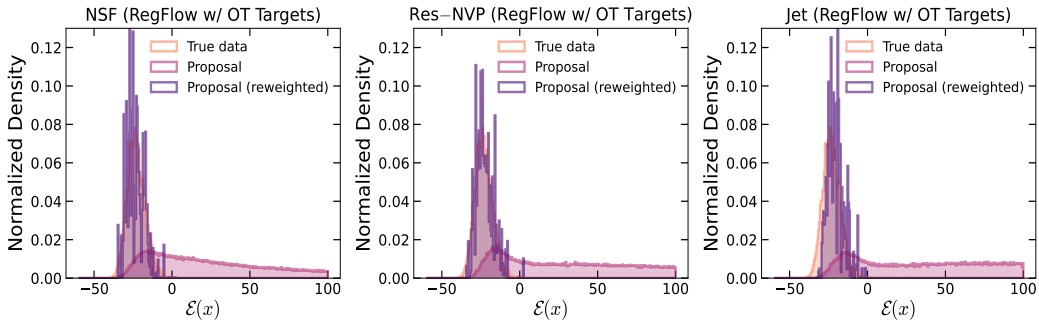

Figure 9: Energy distribution of the original and re-weighted samples, as well as the true data, when using 100,000 OT targets on ALDP (**left**: NSF (REGFLOW); **center**: Res−NVP (REGFLOW); **right**: Jet (REGFLOW)).

# D  ADDITIONAL RESULTS

## D.1  REGFLOW PERFORMANCE USING OT TARGETS

**Optimal transport targets.**  In addition to using reflow targets from a pre-trained CNF, we pre-compute an OT map to obtain an invertible pairing between source and target samples. We combine this map with REGFLOW training, and report results in fig. 9 for alanine dipeptide. Here, we demonstrate an example of where REGFLOW training goes beyond distillation and can serve as an effective approach at training classical normalizing flows on diverse invertible maps.

## D.2  PERFORMANCE ON LARGER PEPTIDES

**Alanine tripeptide and alanine tetrapeptide**  We demonstrate the learned distributions of the two pairs of dihedral angles that parameterize alanine tripeptide and tetrapeptide using our best MLE-trained and REGFLOW flows in fig. 11 and fig. 12. The inability to capture the modes using MLE is elucidated, where multiple modes appear to blend together in both sets of dihedral angles in fig. 11. Conversely, using REGFLOW, most modes are accurately captured and the general form of the Ramachandran plots conforms well to that of the true distribution obtained from MD. The findings observed with alanine tripeptide are even more pronounced with alanine tetrapeptide, where certain modes are entirely missed when MLE-trained flows are used, as seen in fig. 12. With REGFLOW, however, most modes are accurately captured, and the density distribution is in strong agreement with the ground truth data. These findings clearly demonstrate the utility of a regression-based training objective over conventional MLE for applications to equilibrium conformation sampling of peptides.

In fig. 10, we demonstrate that the energy distribution of the re-weighted samples using REGFLOW, which yields a more favourable energy distribution over MLE-trained flows. For the tripeptide, the results are in strong agreement with MD. For the tetrapeptide, the re-weighted samples are superior than their MLE counterparts, but have room for improvement in matching the true energy distribution.

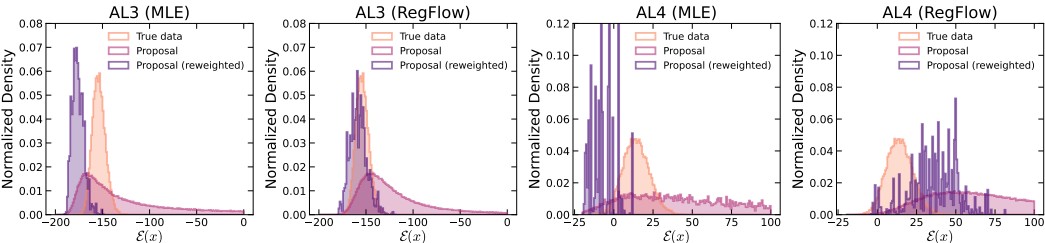

Figure 10: Energy distribution of original and re-weighted samples generated for the most performant MLE and REGFLOW models on alanine tripeptide (**left** and **center left**) and alanine tetrapeptide (**center right** and **right**).

## D.3  GENERATED SAMPLES OF PEPTIDE CONFORMATIONS

**Samples of generated peptides**. Below we provide sample conformations of alanine dipeptide generated using both MLE training and REGFLOW in fig. 13. In addition, we include sample molecules of the larger peptides, obtained through REGFLOW training as well in fig. 14.

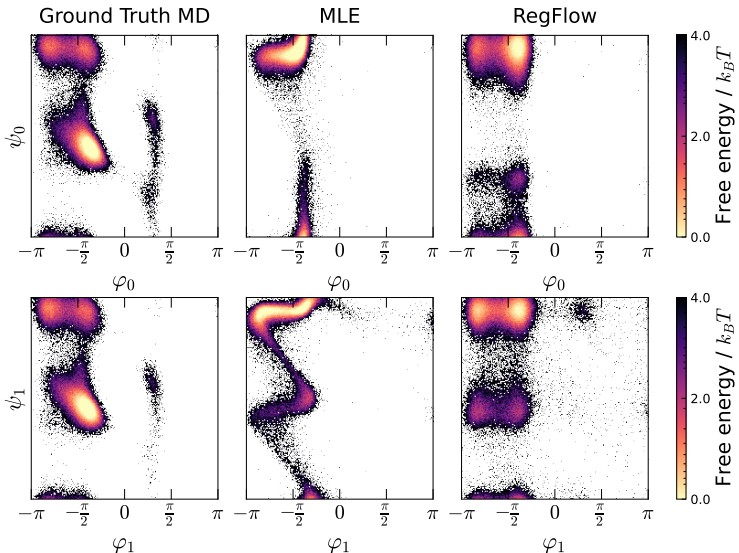

Figure 11: Ramachandran plots for alanine tripeptide (**left**: ground truth, **middle**: best MLE-trained flow, **right**: best REGFLOW flow). REGFLOW captures most modes, while MLE-trained flows struggle.

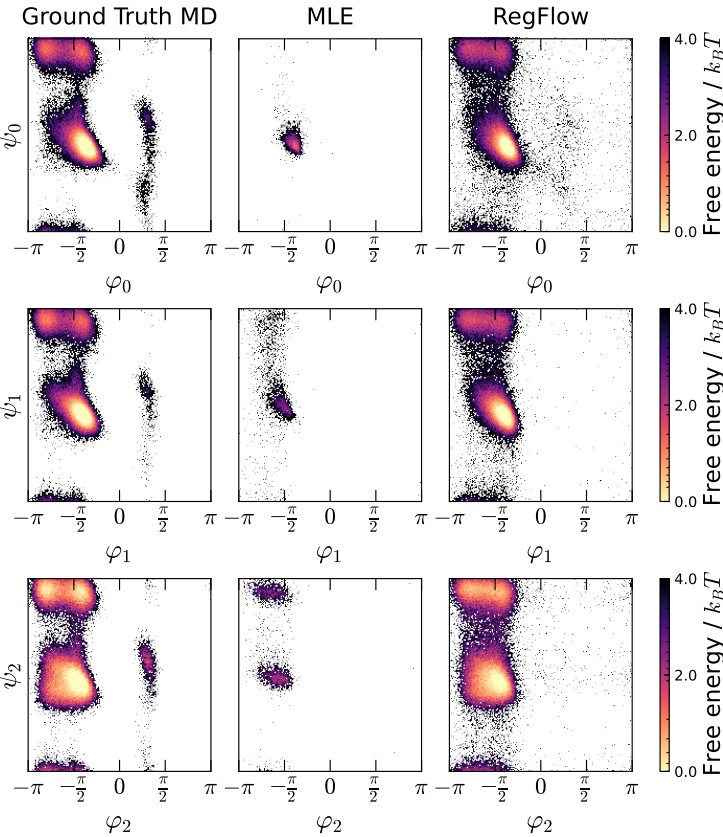

Figure 12: Ramachandran plots for alanine tetrapeptide (**left**: ground truth, **middle**: best MLE-trained flow, **right**: best REGFLOW flow). REGFLOW captures most modes, while MLE-trained flows struggle.

### D.4    TARGETED FREE ENERGY PERTURBATION

**Generating regression targets**. Using the available MD data, two conformations of alanine dipeptide were selected: $\beta_{\text{planar}}$ and $\alpha_{\text{R}}$ (Ghamari et al., 2022). The $(\phi, \psi)$ ranges for the $\beta_{\text{planar}}$ conforma-

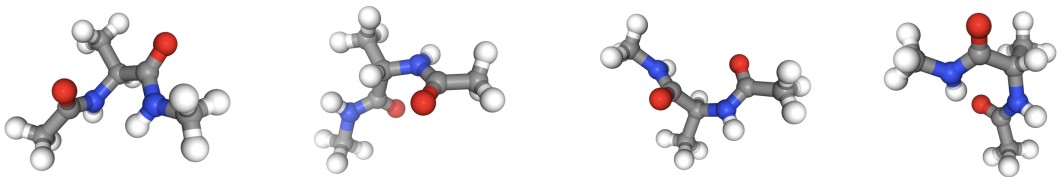

Figure 13: Generated conformations of alanine dipeptide across various flow-based methods (**left**: NSF w/ MLE; **center left**: NSF w/ REGFLOW; **center right**: Res–NVP w/ REGFLOW; **right**: Jet w/ REGFLOW.

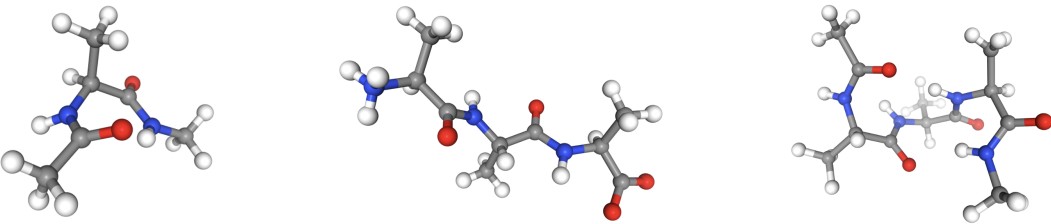

Figure 14: Generated samples of larger peptides using NSF (REGFLOW) (**left**: ALDP; **center**: AL3; **right**: AL4).

tion were chosen as $(-2.5, -2.2)$ and $(2.3, 2.6)$, and for the $\alpha_R$ conformation as $(-1.45, -1.2)$ and $(-0.7, -0.4)$, respectively. The dataset was then truncated to 82,024 source-target conformation pairs, which were used to compute the OT pairing and generate an invertible map. These pairs were subsequently trained using REGFLOW, with the same model configurations and settings outlined in table 9.

