# OpenReview forum: "Efficient Regression-based Training of Normalizing Flows for Boltzmann Generators"
_ICLR.cc/2026/Conference — ICLR 2026 Poster_

### Official Review · Reviewer_dEB8 · 2025-10-20

**Soundness:** 3
**Presentation:** 3
**Contribution:** 3
**Rating:** 6
**Confidence:** 4

**Summary:**

This paper proposes to train normalizing flows by regression to alleviate the training instability in traditional MLE training to transform samples from a prior $q$ to a target distribution $p$, ie $x_0\sim q$ and $x_1=T(x_0)\sim p$. To enable regression-based training, it requires that there exists an invertible map between $x_0$ and $x_1$. Finding an exact solution is intractable, and therefore the author proposes two different approximations: 1. through coupling $(x_0, x_1)$ with optimal transport; 2. through coupling $(x_0, x_1)$ by training an additional CNF and letting $x_1$ be generated from $x_0$ through the trained CNF.

The author justifies the usage of their CNF-coupling by showing the error bound wrt wasserstein distance between the learned distribution and the target distribution. The experimental results also showcase the effectiveness of the proposed method, through various of peptide tasks.

In summary, the reviewer gives a weak acceptance.

**Strengths:**

1. The proposed method is simple and effective, which improves the training stability of NFs

2. The author provides mathematical justification for using the CNF-coupling, by showing the wasserstein distance between trained model and target.

3. Though the CNF-coupling sounds expensive at the first glance, the author shows that it requires much less computational overhead in table 4.

4. The experimental results are good

**Weaknesses:**

1. The main concern lies in the invertibility of coupling, as both OT-coupling and CNF-coupling are approximations. It would be great if the author could elaborate on when this approximation would be broken and, in such a case, how the classic MLE objective would help. Intuitively speaking, if $p_0$ and $p_1$ are too separate, the true velocity might be less smooth and the (t-dependent) Lipschitz constants can be large, which means the Wasserstein bound can be very loose.

2. The experiments in this paper focus on training NFs and then doing importance sampling to get equilibrium samples. However, in the plots, such as Figure 2, the author only shows the reweighted energy histogram but not the resampled Ramachandran plots. The reviewer thinks it is important as well to show the rama-plots with recalibrated samples.

**Questions:**

1. Can the author also show the training time comparison analogous to table 4 on other benchmarks?

2. Table 3 is a bit confusing. Why are the inference times of the same NF but different training methods (MLE/RegFlow) different?

---

> ### Author Response · Authors · 2025-11-23
> **Response to Reviewer dEB8**
>
> We would like to thank the reviewer for their score and confidence in our approach. We appreciate that the reviewer found our RegFlow approach to be “simple and effective”, and that it improves the “training stability of NFs”. We also value that the reviewer agrees that RegFlows “requires much less computational overhead” and that our experimental results “are good”. We now address the key clarifications in the review grouped by theme.
>
>
> ## Invertibility of Coupling
> > 1. The main concern lies in the invertibility of coupling, as both OT-coupling and CNF-coupling are approximations. It would be great if the author could elaborate on when this approximation would be broken and, in such a case, how the classic MLE objective would help.
>
> We believe there is a small misunderstanding in the requirements of the coupling which we would like to clarify here and have clarified in the draft. Any coupling is usable for training RegFlow as long as there exists an invertible function on the continuous domain which agrees with it. This leaves us with easily verifiable necessary and sufficient properties for the base coupling. Specifically, let $\pi(x_0, x_1)$ denote the coupling between empirical point sets $x_0, x_1$ in $\mathbb{R}^d$.
>
> If $\pi$ is a permutation and there does not exist $i \neq j$ such that $x_0^i = x_0^j$ or $x_1^i = x_1^j$. Then there exists invertible function $f: \mathbb{R}^d \to \mathbb{R}^d$ such that for all $x_0^i \in x_0$, $f(x_0^i) = x_1^i)$ and for all $x_1^i \in x_1$ $f^{-1}(x_1^i) = x_0^i$, which establishes necessary and sufficient conditions for training a valid RegFlow between $x_0$ and $x_1$. This is actually quite easy to obtain in practice. For both OT-couplings and CNF-couplings, $\pi$ is almost by definition a permutation. The only trouble is if there exist duplicate points. This is a measure-zero event in continuous space, and therefore is not an issue.
>
> The more interesting question perhaps, is if $\pi$ is a “good” coupling in the sense that it is somehow “easy” to learn and generalizes well when trained with the RegFlow procedure in a given setting. While a random permutation between (non repeating) samples of the data and Gaussian is technically valid, it is fairly clear that this is not a “good” coupling.
>
> We believe the classic MLE objective would help when it is difficult to find a good coupling for the given architecture, dataset, and RegFlow learning framework. In this work we established that there exist settings where OT and CNF-coupings outperform the MLE objective. We leave it to future work to study the optimal couplings in a given setting. We hope this clarifies the nature of the requirements of the coupling for RegFlow and when the MLE objective may help. We refer the reviewer to Section A.4 of the updated draft.
>
> ## Ramachandran plots with reweighting
>
> > 2.  In the plots, such as Figure 2, the authors only show the reweighted energy histogram but not the resampled Ramachandran plots.
>
> We thank the reviewer for their comment and apologize for any confusion—in Fig. 2, we do show both the energy distribution as well as the resampled Ramachandran plots (the fourth and fifth plot specifically, while the third plot corresponds to the ground truth MD Rama). The fourth and fifth plots demonstrate the resampled torsion angle distributions for our most performant MLE-trained flow and most performant RegFlow model. Here—as the reviewer very correctly describes as important—we demonstrate the merging and missing of conformational modes by the MLE-trained flow, while the RegFlow-trained NF demonstrates the distinct conformation modes, as well as a correctly sampled last mode which is largely ignored by the same model trained with MLE. The reviewer is entirely correct about how the Ramachandran plots are essential at demonstrating the correctness of local features in the learned model, and here we note that our approach not only outperforms MLE on global metrics, but on local metrics as well, as shown in the resampled Ramachandran plots. We also include the same plot types for the larger peptides—alanine tripeptide and alanine tetrapeptide—in the appendix, in Fig. 10 and 11 given its importance in demonstrating the efficacy of the RegFlow training process. Lastly, since it may not have been clear from the caption that the Ramachandran plots are of the resampled points (and not the proposal), we have clarified in the revised manuscript to prevent any confusion.

---

> > ### Author Response · Authors · 2025-11-23
> > **Response to Reviewer dEB8 (Continued)**
> >
> > ## Questions
> >
> > > 3. Can the author also show the training time comparison analogous to table 4 on other benchmarks?
> >
> > We would like to thank the reviewer for their comment around improved benchmarking of RegFlow in contrast to MLE for the other systems considered. In line with this comment, we have summarized these results across all peptides through a new Figure that we add to the main text–Fig. 3. Fig. 3 demonstrates the time it takes for RegFlow to outperform MLE during training using the most performant MLE models (NSF). We plot the energy-W1 as a function of training time and include dotted lines to illustrate the best energy-W1’s obtained during the MLE training runs. Here, we can see that the crossover occurs at ~1h20, ~1h16, and ~2h45, for RegFlow to outperform MLE on the dipeptide, tripeptide, and tetrapeptide, respectively, when considering the energy-W1 metric. In line with these learnings, we have included additional supporting information in the main text.
> >
> >
> > > 4. Why are the inference times of the same NF different depending on how they’re trained?
> >
> > This is an excellent question that also demonstrates an additional benefit of the RegFlow training process. If we consider the neural spline flow (NSF): an autoregressive normalizing flow, the forward direction (data $\to$ latents) is far faster to compute than the inverse (latents $\to$ data). Consequently, during MLE training—due to this asymmetric computational cost—the loss is computed in the forward (fast) direction due to the frequency at which it is called. Then, after the model has been trained, inference is slow, as the reverse direction needs to be called to generate samples and compute likelihoods. If for instance, we train using RegFlow and leverage the log determinant as a regularizer, the loss function—similar to the MLE objective—only uses the forward (fast) direction, yet this time to go from latents $\to$ data. Then, during inference, since the flow has already been parameterized to go from latents $\to$ data, **inference does not need to be slow.** The benefit of RegFlows here is that if the user aims to train a normalizing flow that has an asymmetric compute cost, they have the flexibility of choosing which direction to parameterize it to optimize for speed. That being said, although this is true for normalizing flows with an asymmetric compute cost, those with analytical inverses—like Res-NVP or Jet—do not benefit from this advantage, as both the forward and backward passes are essentially the same speed.
> >
> > ### Closing comments
> > Overall, we would like to thank the reviewer for their detailed and insightful comments, and greatly appreciate their dedicated time and effort to reviewing our work. We hope that through these additional clarifications and modifications, we have done an effective job at communicating the potential for RegFlow training to enable more efficient and scalable Boltzmann Generation, leading the reviewer to potentially consider increasing their score.

---

> > > ### Comment · Reviewer_dEB8 · 2025-11-26
> > > **Response to author**
> > >
> > > The reviewer thanks the author's responses which address all raised concerns. The reviewer would increase the score.

---

### Official Review · Reviewer_fKR3 · 2025-10-26

**Soundness:** 4
**Presentation:** 4
**Contribution:** 3
**Rating:** 6
**Confidence:** 4

**Summary:**

This manuscript outlines a new approach to train one-step generative models, specifically normalizing flows, which allow exact sample likelihood evaluations. This work is important as likelihood evaluation is a bottleneck for applications of NF in the sciences, as most prominently outlined here: Boltzmann Generators. The idea outlined in this paper is simple and effective: train a regular NF against a pre-specified flow (e.g. either a pre-trained Continuous NF, or a pre-computed OT flow). This allows for faster and more stable training and more efficient sample likelihood evaluations in most cases.

While the method seems like it still has some ways to go to be ready for prime-time, I find that the paper, overall, is an interesting conceptual step. Consequently, i am willing to increase my score if the concerns below are addressed.

**Strengths:**

- Conceptually clear and well written manuscript. Numerous insightful comments about normalizing flows.
- Well thought-out experiments and evaluations. All fairly standard in the field now, but still well done.
- Clear performance gain, in terms of compute-time, over most of the included baselines.

**Weaknesses:**

- Claims and attribution. The proposed TFEP method is closely related to the ambient thermodynamic interpolant approach by Moqvist et al https://arxiv.org/abs/2411.10075
- Sample quality and scaling. ESS remain fairly low. Scaling to tetra peptides is nice several other recent works e.g. https://arxiv.org/abs/2502.18462 demonstrate scaling to significantly larger systems.
- Lack of error estimates on evaluation statistics.

**Questions:**

- How do the authors envision this approach scale to larger systems? Comparing figures 7 and 8 it seems like the RegFlows miss important details that might be important for performance, and one would expect this to only increase with system size.
- Are observables --- e.g. free energies --- computed under the DiT-based CNF meaningfully different from those under the reweighed values from the presented RegFlow approach?

---

> ### Author Response · Authors · 2025-11-23
> **Response to Reviewer fKR3**
>
> We would like to thank the reviewer for their thorough review and consideration of our work. We are grateful that the reviewer found our work to be “conceptually clear and well written”, and that it provided “numerous insightful comments about normalizing flows”. We are also appreciative that the reviewer also found our experiments to be well thought out and there is a “clear performance gain, in terms of compute-time”. We next comprehensively address all the key points raised in the review.
>
>
> ## Claims and Attribution
>
> > 1. The proposed TFEP method is closely related to the ambient thermodynamic interpolant approach by Moqvist et al.
>
> We thank the reviewer for sharing this work. We have carefully reviewed the TFEP approach of Moqvist et al. Here, we note that although the method evaluates the TFEP in a similar manner, the models used to achieve the result are entirely different – Moqvist et al. illustrate how continuous normalizing flows (CNFs) can be used to learn invertible maps between two target states. In contrast, our approach demonstrates two things: (1) that this can be done in one-step using a discrete normalizing flow; and (2) pairing of samples between two conformational states do not need to be obtained from a CNF–but instead–through solving the OT problem, followed by regression training to obtain the TFEP between two meta-stable states. Thus, TFEP and RegFlows in this setting have similar goals but dramatically different algorithmic approaches. We thank the reviewer again for pointing out the related work; we have cited the work in the updated PDF allowed in this rebuttal.
>
> ## Sample quality and scaling
>
> > 2. Sample quality and scaling. ESS remains fairly low. Better scaling to tetrapeptides has been seen in other recent work by Tan et al.
>
> We thank the reviewer for their attention to detail. Although the performance across metrics is reduced relative to [1], the focus of this work was illustrating that a significant number of previously discarded architectures—that fall prey to the instability of the MLE objective—can be revived through a modified training procedure. More precisely, RegFlows  instantiate the same models, with the same number of parameters, but an alternate training method to unlock scalable performance to larger peptide systems. Through the RegFlows training process, we show that invertible architectures can scale to larger peptides and that, with further advances in more expressive and powerful models with larger capacities, they may ultimately support further scaling to even longer peptides and, eventually, full proteins. Consequently, the central claim in our paper is to demonstrate the improvement over MLE for Boltzmann Generators for which we feel the current crop of experiments thoroughly demonstrate on 3 distinct, but well-known normalizing flow architectures.
>
> > 3. Lack of error estimates on evaluation statistics.
>
> We appreciate the reviewer’s dedication to accuracy. As a result, we have trained separate models, each using three seeds, and updated the results in Table 2 to reflect these updates. The results reported correspond to those from the forward-backward regularization strategy, which entirely eliminates the need for the log determinants during training. The table is reproduced below for ease of reference.
>
> | Algorithm ↓ | **ALDP (Dipeptide)** | | | **AL3 (Tripeptide)** | | | **AL4 (Tetrapeptide)** | | |
> |-------------|-----------------------|---|---|-----------------------|---|---|------------------------|---|---|
> |             | ESS ↑ | E-W₁ ↓ | T-W₂ ↓ | ESS ↑ | E-W₁ ↓ | T-W₂ ↓ | ESS ↑ | E-W₁ ↓ | T-W₂ ↓ |
> | NSF (MLE) | **0.055±0.012** | 13.797±2.713 | 1.243±0.103 | 0.024±0.004 | 17.596±1.21 | 1.665±0.180 | **0.016±0.003** | 20.886±1.93 | 3.885±0.41 |
> | NSF (RegFlow) | 0.035±0.004 | **0.501±0.011** | **0.951±0.054** | **0.031±0.018** | **0.853±0.105** | **1.577±0.140** | 0.011±0.003 | **3.277±0.546** | **2.342±0.102** |
> | Res-NVP (MLE) | <1e-4 | >1e3 | >30 | <1e-4 | >1e3 | >30 | <1e-4 | >1e3 | >30 |
> | Res-NVP (RegFlow) | **0.035±0.008** | **2.104±0.586** | **0.812±0.121** | **0.025±0.006** | **3.241±0.301** | **1.881±0.205** | **0.013±0.004** | **2.705±0.306** | **2.117±0.331** |
> | Jet (MLE) | <1e-4 | >1e3 | >30 | <1e-4 | >1e3 | >30 | <1e-4 | >1e3 | >30 |
> | Jet (RegFlow) | **0.055±0.006** | **4.193±1.016** | **0.801±0.076** | <1e-4 | >1e3 | **3.644±0.358** | <1e-4 | >1e3 | >30 |

---

> > ### Author Response · Authors · 2025-11-23
> > **Response to Reviewer fKR3 (Continued)**
> >
> > Further, in our ablation contrasting regularization strategies and target choice, we also trained separate models using three separate seeds, and consolidated the findings into an updated Table 5. Again, we include the table below for ease of reference.
> >
> > | Model                          | ESS           | E-W₁           | T-W₂           |
> > |--------------------------------|---------------|----------------|----------------|
> > | NSF (MLE)                      | **0.055±0.012** | 13.797±2.713   | 1.243±0.103    |
> > | NSF (FORT w/ logdets)          | 0.036±0.007   | 0.519±0.021    | 0.958±0.074    |
> > | NSF (FORT w/o reg)             | 0.032±0.008   | 0.604±0.045    | 1.083±0.109    |
> > | NSF (FORT w/ fwd-bwd)          | 0.035±0.004   | **0.501±0.011** | **0.951±0.054** |
> > | Res-NVP (MLE)                  | <1e-4         | >1e3           | >30            |
> > | Res-NVP (FORT w/ logdets)      | 0.032±0.008   | 2.310±0.411    | 0.796±0.109    |
> > | Res-NVP (FORT w/o reg)         | 0.033±0.010   | 2.948±0.457    | 1.179±0.218    |
> > | Res-NVP (FORT w/ fwd-bwd)      | **0.035±0.008**   | **2.104±0.586** | **0.812±0.121** |
> > | Jet (MLE)                      | <1e-4         | >1e3           | >30            |
> > | Jet (FORT w/ logdets)          | 0.051±0.004   | 6.349±1.412    | 0.872±0.065    |
> > | Jet (FORT w/o reg)             | 0.053±0.007   | 9.707±1.843    | 1.224±0.181    |
> > | Jet (FORT w/ fwd-bwd)          | **0.055±0.006** | **4.193±1.016**    | **0.801±0.076** |
> >
> > > 4. How do the authors envision this approach scale to larger systems? Comparing figures 7 and 8 it seems like the RegFlows miss important details that might be important for performance, and one would expect this to only increase with system size.
> >
> > We thank the reviewer for their points regarding the scaling of RegFlows—this represents an important question in the practical utility of our proposed approach. We would like to politely state that prior to RegFlows, scaling neural spline flows, RealNVPs, or even state-of-the-art discrete NFs like Jet beyond alanine dipeptide was not possible using MLE training. RegFlows, through only modifications to the training process, recasts the problem into one that serves to unlock these architectures for such applications. Although we have demonstrated a proof of concept to systems as large as alanine tetrapeptide, our hopes are twofold: (1) through more effective recent advancements in CNFs or more efficient approaches to evaluating OT maps [3], we can gain access to higher quality invertible mappings that provide RegFlows with an outstanding starting point for learning complex systems; and (2) improved discrete NF architectures—with more powerful and expressive potential—may more effectively unlock the true potential for RegFlows. In principle, a combination of these efforts could be where we see the true scaling capability of RegFlows to longer peptides, and eventually proteins, is achieved. This is a ripe direction for future work.
> >
> > > 5. Are observable, computed under the DiT-based CNF, meaningfully different from those under the reweighted values from the presented RegFlow approach.
> >
> > The reviewer makes an excellent point: one can directly train a CNF on samples from the two metastable states and use the model’s learned likelihoods to perform TFEP, providing an additional comparison point for RegFlow. In line with this, we took the same set of samples from the two defined meta-stables states, and trained the DiT CNF on learning to map from one state to another. After training the CNF, we employed a similar approach to that described in the paper, to determine the desired observable–the free energy difference. We obtained a result that was in close agreement with that obtained from the MD simulations, along with those obtained via RegFlows. However, the total time taken to compute the free energy differences was **two orders of magnitude longer than that taken when using RegFlows.** We added the result from the trained CNF to Fig. 4, as an additional baseline, and have included text in the main body of the paper to summarize these findings. Overall, we thank the reviewer for their excellent suggestion, and believe that this has improved the quality of our manuscript.
> >
> > ### Closing comments
> > Overall, we thank the reviewer for their insightful comments and suggestions—we hope that our rigorous discussion, and additional experiments and clarifications lead the reviewer to reconsider their score, and improve their rating, based on their original comments and our rebuttal in context.
> >
> > ### References
> >
> > [1] Tan, Charlie B., et al. "Scalable equilibrium sampling with sequential boltzmann generators." arXiv preprint arXiv:2502.18462 (2025).
> >
> > [2] Zheng, Guangting, et al. "FARMER: Flow AutoRegressive Transformer over Pixels." arXiv preprint arXiv:2510.23588 (2025).
> >
> > [3] Halmos, Peter, et al. "Hierarchical Refinement: Optimal Transport to Infinity and Beyond." arXiv preprint arXiv:2503.03025 (2025).

---

> > > ### Comment · Reviewer_fKR3 · 2025-11-23
> > >
> > > Thanks for addressing all of my comments. I'll increase my score.

---

### Official Review · Reviewer_Upe7 · 2025-11-01

**Soundness:** 4
**Presentation:** 4
**Contribution:** 4
**Rating:** 8
**Confidence:** 4

**Summary:**

This paper describes REGFLOW, an approach for training a normalizing flow that performs better than traditional maximum likelihood training.  The approach regresses to predetermined flows, either from another model or precomputed optimal transport couplings. This approach results in substantial better normalizing flow models than traditional training.

**Strengths:**

The paper addresses a chronic problem with normalizing flows.  The described regression loss is intuitive an simple to implement (once couplings have been determined).  The approach results in dramatic improvements compared to MLE training using the same models and data.  Sensitivity to some parameters (e.g. regularization) is explored. I appreciate the evaluation of free energy.  The paper is well written and easy to follow.

**Weaknesses:**

Although the improvement compared to NF models is extreme, the results aren't necessarily state-of-the-art compared to other models.

**Questions:**

Why are NFs trained with REGFLOW substantially faster at computing likelihoods?

How does increasing the number of OT couplings improve performance? What if the OT is approximate?

What is the basis for the statement that beyond a certain level of regularization that numerical invertability is guaranteed? Is this an empirical statement, or is there are proof (if the former, than perhaps an alternate wording would be more accurate).

---

> ### Author Response · Authors · 2025-11-23
> **Response to Reviewer Upe7**
>
> We would like to thank the reviewer for their time and effort, as well as their positive appreciation of our work. We are delighted to hear that the reviewer agrees that our paper addresses a “chronic” problem with normalizing flows and that our proposed RegFlows offer an “intuitive and simple to implement” solution. We are also heartened to hear that the reviewer considers the demonstrated results to show “dramatic improvements compared to MLE”—which we highlighted as a central contribution of this work. Finally, we thank the reviewer for acknowledging that our paper is “well written and easy to follow”. We next address the main clarification points from the review, grouped by theme.
>
> ### Results in comparison to state-of-the-art models
>
> We agree with the reviewer that while RegFlows outperform MLE on identical architectures, the absolute performance does not always surpass specialized, SOTA models (which often rely on specific domain constraints). Our primary goal was to benchmark the *training method* (RegFlow vs. MLE) in a controlled setting, rather than perform an architecture search. By fixing the architecture (e.g., NSF, Res-NVP, Jet) and data, we isolate the contribution of the training objective. As noted in the paper, MLE training on these standard architectures often failed on larger peptide systems, whereas RegFlow training succeeded. More precisely, for the architectures considered: NSF, Res-NVP (RealNVP with a Resnet parameterization), and Jet, RegFlows are a state-of-the-art method, without making further claims of generally being state-of-the-art on the benchmark. This suggests that RegFlows could potentially enable SOTA architectures to achieve even better results, though we leave the investigation of RegFlows on other architectures for future work.
>
> ### Questions
> > 1. Why are NFs trained with RegFlow substantially faster at computing likelihoods?
>
> When comparing MLE vs. RegFlow-trained discrete flows, we also find that for flows with an analytic determinant—like with RealNVPs or Jet—RegFlow-trained flows are slightly faster (but mostly comparable) to MLE. However, for autoregressive flows like NSF, where one direction of the flow is typically slower than the other, RegFlow-trained flows can be up to 35x faster than MLE-trained flows. If the MLE flow were trained in the opposite direction, then inference would speed up, but training would be far slower. We have added additional text to clarify this reason and thank the reviewer for this opportunity to clarify the benefits of our work.
>
> > 2. How does increasing the number of OT couplings improve performance? What if the OT is approximate?
>
> We appreciate the reviewer's question. We answer by first noting that, as the training data used for MLE leverages 100,000 samples from the Markov chain, to enable a fair comparison, we chose to use the same number of points for computing OT targets.
>
> For most approximate OT methods (e.g. entropic or minibatch regularized), the coupling is no longer invertible and is therefore unsuitable for our application. Technically, any hard coupling (permutation) of empirical distributions defines an invertible diffeomorphism, so even a random permutation could work. However, clearly this coupling may be difficult to learn. In this paper we demonstrate that different coupling types can be used, but do not yet have guidance on what kinds of couplings are easiest to learn in general as this will depend both heavily on the model and data. We add discussion of this and theoretical requirements of the coupling in section A.4 of the appendix.
>
> > 3. What is the basis for the statement that beyond a certain level of regularization that numerical invertibility is guaranteed?
>
> This statement was an empirical statement based on our observations tracking the empirical invertibility. Specifically, during training, we track how invertible the network is by tracking $\mathcal{L}_{fwd-bwd}$ on a validation set. We have modified the text to clarify this.
>
> ### Closing comments
>
> We thank the reviewer for their time and effort in reviewing this paper. We believe we have answered all the great points raised by the reviewer in our rebuttal, and we remain eager to help clarify any additional questions if they arise.
>
> ### References
> [1] Kolesnikov, Alexander, André Susano Pinto, and Michael Tschannen. "Jet: A modern transformer-based normalizing flow." arXiv preprint arXiv:2412.15129 (2024).

---

### Author Response · Authors · 2025-11-23
**Global Response**

We sincerely thank all the reviewers for their time, energy, and thoughtful feedback. Across the reviews, we are grateful that reviewers consistently noted that RegFlows are “intuitive and simple to implement”, and provide an “effective” way to improve the training stability of normalizing flows via our novel regression objective (Upe7, fKR3, dEB8). We also appreciated that the reviewers found our paper to be well-written, conceptually clear, and provided insightful commentary on the challenges of normalizing flow training (UPe7, fKR3). We also greatly value that reviewer fKR3 and reviewer dEB8 noted that the experiments are well-designed and empirically compelling, while reviewer Upe7 specifically praised the “dramatic improvements” over MLE on identical architectures. This was further recognized amongst reviewers who noted that RegFlows provides substantial computational benefits, including faster and more stable likelihood evaluations, reduced overhead compared to CNF-based approaches, and the ability to scale previously unstable NF architectures to larger peptide systems.

We now turn our attention to the shared concerns raised by the reviewers in this global response, before tackling specific comments to each reviewer in their individual responses. We also use this response as an opportunity to summarize the key new experiments included as part of our rebuttal. All revisions in our updated draft are highlighted in **blue** for readability.

## Computational speedups of RegFlows (Reviewers Upe7, dEB8)

Reviewers Upe7 and dEB8 asked for clarification on why inference time was substantially faster for RegFlow as compared to MLE for some architectures. This is because RegFlows allows us to train the network in the same direction that inference is performed. Some architectures are substantially faster in one direction than the other. In these cases, we can use the fast direction both for training and for inference. For MLE training the fast direction is always used for training, and therefore causes a slowdown during inference relative to RegFlow training of the same architecture. We have added a paragraph in section 3.1 to clarify this.
## Quality of couplings (Reviewers Upe7, dEB8)
Reviewers Upe7 and dEB8 asked about the quality of the couplings used for RegFlow. To clarify this we have added section A.4 of the appendix discussing the theoretical requirements for the coupling and guidance on what may be useful in investigation of other couplings.
## New Experiments

### DiT CNF for TFEP
Reviewer fKR3 highlighted their desire to better understand the performance-inference gap between the free energies computed by our DiT CNF vs. the approach taken by RegFlow. This approach circumvents the need for computing the OT map between samples from both the meta-stable states; however, adds the additional challenge of needing to compute the likelihoods to determine the TFEP. To demonstrate this, we ran an additional experiment using the same data that was used to compute the OT targets, but this time to train the DiT CNF. Then, we computed the TFEP using the trained CNF, but obtained the likelihoods through direct integration of the learned vector field—the most time consuming step. We updated Fig. 4 with the estimates of the free energy difference obtained using the CNF, which are in close alignment with those obtained through the MD simulations along with RegFlow; however, to obtain the TFEP using the DiT CNF, the time taken was nearly two orders of magnitude larger than that of RegFlow, demonstrating the efficiency of the regression training process, without significant drops in accuracy.

### Training speed comparison on larger peptides
As suggested by reviewer dEB8, we introduce a new figure (Fig. 3) demonstrating training speed of RegFlow compared to MLE on larger peptides. This demonstrates that NFs trained with RegFlow are able to achieve the same performance as an MLE trained model in significantly less time.

### Additional model seeds
As highlighted by reviewer fKR3, there was a lack of error estimates for the reported results on all peptides considered. We have additionally trained models across three separate seeds for all peptides under consideration, and updated our main results in Table 2 and 5, in accordance with these simulations. In this process, we can still clearly see the superior performance of RegFlow over MLE training across the architectures studied across both global and local metrics.

We strongly believe these changes make the paper clearer and more complete, and we thank the reviewers again for helping us strengthen it.

---

### Author Response · Authors · 2025-12-03
**Final Remarks**

We would like to thank the AC for their effort in handling our paper—especially during this year’s more burdensome review process. We would also like to sincerely thank the reviewers for the time and care they have dedicated to rigorously evaluating our work. Their insights were extremely valuable and have meaningfully strengthened the paper. We also wish to express our appreciation to both the previous and current ACs for their careful consideration of the reviews and our rebuttal. Our paper considers a fundamentally new way of training classical Normalizing Flows for Boltzmann Generators using a regression objective—compared to standard MLE training—unlocking new, more efficient training and scaling on architectures that were previously discarded in the literature. We would like to summarize the discussion as well as how we addressed all reviewer concerns.

- Reviewer Upe7: Initial score 8. Unfortunately, we received no response before the discussion freeze. This reviewer was extremely positive of the work with soundness, presentation, and contribution all scored excellent in their initial review, with minor concerns which we addressed in our response.
- Reviewer fKR3: Initial score 6 → 8 post discussion confirming all comments were addressed sufficiently.
- Reviewer dEB8: Initial score 6 → 8 post discussion confirming that the author’s responses addressed all raised concerns.

We are confident that the already positive assessments, together with the additional experiments conducted during the rebuttal phase, underscore the contribution and robustness of our work.

During the rebuttal period, we addressed the reviewers’ shared concerns and introduced several targeted experiments and clarifications that directly contributed to the shift in assessments. All updates in the revised draft are highlighted in **blue** for clarity.

- Reviewers Upe7 and dEB8 inquired about the computational speedups observed for RegFlows; in response, we expanded Section 3.1 to explain how RegFlows allow training in the same direction as inference, enabling certain architectures to exploit their inherently faster direction for both training and sampling—an advantage not available to MLE-based training.
- We added Appendix Section A.4 to clarify the theoretical requirements and practical considerations for couplings used in RegFlow, addressing questions about coupling quality raised by the same reviewers.

Beyond these clarifications, we performed three key new experiments that sufficiently addressed all remaining concerns of the reviewers.

- To respond to reviewer fKR3’s request to better understand the performance–inference gap between DiT CNF and RegFlow, we trained a DiT CNF on the same dataset used for OT target construction and computed TFEP estimates through direct integration of the learned vector field. As shown in the updated Fig. 4, the resulting free energies align closely with MD simulations and RegFlow, but require nearly two orders of magnitude more computation time—highlighting RegFlow’s efficiency without sacrificing accuracy.
- Following reviewer dEB8’s suggestion, we added a new experiment on larger peptides (Fig. 3) demonstrating that RegFlow achieves MLE-level performance in significantly less training time.
- In response to reviewer fKR3’s request for error estimates, we trained all models across three independent seeds and updated Tables 2 and 5 accordingly; these results reaffirm the consistent superiority of RegFlow across architectures and metrics.

Collectively, these additions not only resolved all outstanding concerns but also strengthened the empirical and theoretical foundations of the work, ultimately contributing to the improved reviewer scores. We believe the revised paper is both rigorous and impactful, and are confident that it makes a strong and valuable contribution to the scientific community by unlocking new scalable training methods for classical Normalizing Flows.

---

### Meta-Review · Area_Chair_mhxw · 2026-01-06

**Summary:**

The paper introduces "RegFlow," a novel training framework designed to overcome the instability and computational bottlenecks associated with Maximum Likelihood Estimation (MLE) training of Normalizing Flows, specifically within the context of Boltzmann Generators for molecular conformation sampling. The authors propose replacing the standard MLE objective with a simpler $\ell_2$-regression loss. To enable this, they construct target pairings using either Optimal Transport (OT) couplings or a pre-trained Continuous Normalizing Flow (CNF). The central claim is that this approach unlocks classical invertible architectures (such as RealNVP and Neural Spline Flows) that were previously difficult to train on molecular systems, offering faster inference and better stability than their MLE counterparts.

During the review process, the most significant concerns identified revolved around the rigor of the empirical evaluation and the theoretical justification for the proposed couplings. While reviewers appreciated the method's simplicity, determining whether RegFlow is truly superior to MLE required more robust evidence than initially presented. Specifically, the initial lack of error estimates (multiple seeds) made it difficult to judge statistical significance. Furthermore, establishing the efficiency gains required clearer benchmarks on training speed and scalability to larger systems (e.g., larger peptides) to prove the method's practical utility over standard baselines. Finally, the theoretical basis for using approximate couplings (and whether they guarantee invertibility) was a critical point of scrutiny that needed clarification to ensure the method's soundness.

I recommend accepting this paper. The authors provided an exemplary rebuttal that directly addressed these identified concerns. The addition of multi-seed error estimates, new experiments on larger peptide systems demonstrating faster convergence than MLE, and theoretical clarifications regarding coupling requirements significantly strengthened the manuscript. By successfully reviving "older" architectures that previously failed on these tasks, the paper offers a practical and efficient tool for the "AI for Science" community. The unanimous consensus among active reviewers following the rebuttal reflects the work's solid methodological contribution and clear empirical utility.

**Reviewer Concerns:**

The rebuttal addressed all substantive concerns. No major issues remain outstanding. In particular, it addressed the following concerns:

* Error Estimates (by reviewer fKR3). Authors re-ran experiments with 3 seeds, updating Tables 2 and 5 to show consistent performance.
* Scaling & Speed (by reviewers dEB8 and fKR3). A new experiment (Fig. 3) demonstrated that RegFlow converges significantly faster than MLE on larger peptides (Alanine Tri/Tetrapeptide).
* Theoretical Justification (by reviewers dEB8 and Upe7). Appendix A.4 was added to clarify that a permutation coupling without duplicates suffices for invertibility.
* Baselines (by reviewer fKR3). Authors compared RegFlow to a DiT-based CNF for Targeted Free Energy Perturbation (TFEP), showing comparable accuracy with ~100x less compute.
* Inference Speed (by reviewer Upe7). Clarified that RegFlow allows training asymmetric flows (like NSF) in their "fast" direction, whereas MLE forces training in the "slow" direction, impacting inference.

**Reviewer Scores:**

* **Reviewer Upe7. Original score: 8. Predicted Score: 8**. The author's response clarified their specific questions on inference speed. As they were already highly positive, the score would arguably stay at 8.
* **Reviewer fKR3. Original score: 6. Predicted Score: 8**. Explicitly stated in the discussion: "Thanks for addressing all of my comments. I'll increase my score." The addition of error bars and TFEP baselines directly addressed their main weaknesses.
* **Reviewer dEB8. Original score: 6. Predicted Score: 8**. Explicitly stated: "The reviewer thanks the author's responses which address all raised concerns. The reviewer would increase the score."
* **Reviewer 57dm**. No review.

---

### Decision · Program_Chairs · 2026-01-26

Accept (Poster)